# A Deep Instance Generative Framework for MILP Solvers Under Limited Data Availability

**Zijie Geng[1], Xijun Li[1,2], Jie Wang[1,3]\*, Xiao Li[1], Yongdong Zhang[1], Feng Wu[1]**
[1]University of Science and Technology of China       [2]Noah's Ark Lab, Huawei
[3]Institute of Artificial Intelligence, Hefei Comprehensive National Science Center
{ustcgzj,lixijun,xiao_li}@mail.ustc.edu.cn
{jiewangx,zhyd73,fengwu}@ustc.edu.cn

## Abstract

In the past few years, there has been an explosive surge in the use of machine learning (ML) techniques to address combinatorial optimization (CO) problems, especially mixed-integer linear programs (MILPs). Despite the achievements, the limited availability of real-world instances often leads to sub-optimal decisions and biased solver assessments, which motivates a suite of synthetic MILP instance generation techniques. However, existing methods either rely heavily on expert-designed formulations or struggle to capture the rich features of real-world instances. To tackle this problem, we propose G2MILP, *the first* deep generative framework for MILP instances. Specifically, G2MILP represents MILP instances as bipartite graphs, and applies a masked variational autoencoder to iteratively corrupt and replace parts of the original graphs to generate new ones. The appealing feature of G2MILP is that it can learn to generate novel and realistic MILP instances without prior expert-designed formulations, while preserving the structures and computational hardness of real-world datasets, simultaneously. Thus the generated instances can facilitate downstream tasks for enhancing MILP solvers under limited data availability. We design a suite of benchmarks to evaluate the quality of the generated MILP instances. Experiments demonstrate that our method can produce instances that closely resemble real-world datasets in terms of both structures and computational hardness. The deliverables are released at https://miralab-ustc.github.io/L2O-G2MILP.

## 1 Introduction

Mixed-integer linear programming (MILP)—a powerful and versatile modeling technique for many real-world problems—lies at the core of combinatorial optimization (CO) research and is widely adopted in various industrial optimization scenarios, such as scheduling [1], planning [2], and portfolio [3]. While MILPs are $\mathcal{NP}$-hard problems [4], machine learning (ML) techniques have recently emerged as a powerful approach for either solving them directly or assisting the solving process [5, 6]. Notable successes include [7] for node selection, [8] for branching decision, and [9] for cut selection, etc.

Despite the achievements, the limited availability of real-world instances, due to labor-intensive data collection and proprietary issues, remains a critical challenge to the research community [5, 10, 11]. Developing practical MILP solvers usually requires as many instances as possible to identify issues through white-box testing [12]. Moreover, machine learning methods for improving MILP solvers often suffer from sub-optimal decisions and biased assessments under limited data availability, thus

---

\*Corresponding author.

37th Conference on Neural Information Processing Systems (NeurIPS 2023).

compromising their generalization to unseen problems [13]. These challenges motivate a suite of synthetic MILP instance generation techniques, which fall into two categories. Some methods rely heavily on expert-designed formulations for specific problems, such as Traveling Salesman Problems (TSPs) [14] or Set Covering problems [15]. However, these methods cannot cover real-world applications where domain-specific expertise or access to the combinatorial structures, due to proprietary issues, is limited. Other methods construct general MILP instances by sampling from an encoding space that controls a few specific statistics [16]. However, these methods often struggle to capture the rich features and the underlying combinatorial structures, resulting in an unsatisfactory alignment with real-world instances.

Developing a deep learning (DL)-based MILP instance generator is a promising approach to address this challenge. Such a generator can actively learn from real-world instances and generate new ones without expert-designed formulations. The generated instances can simulate realistic scenarios, cover more cases, significantly enrich the datasets, and thereby enhance the development of MILP solvers at a relatively low cost. Moreover, this approach has promising technical prospects for understanding the problem space, searching for challenging cases, and learning representations, which we will discuss further in Section 5. While similar techniques have been widely studied for Boolean satisfiability (SAT) problems [17], the development of DL-based MILP instance generators remains a complete blank due to higher technical difficulties, i.e., it involves not only the intrinsic combinatorial structure preservation but also high-precision numerical prediction. This paper aims to lay the foundation for the development of such generators and further empower MILP solver development under limited data availability.

In this paper, we propose G2MILP, which is *the first* deep generative framework for MILP instances. We represent MILP instances as weighted bipartite graphs, where variables and constraints are vertices, and non-zero coefficients are edges. With this representation, we can use graph neural networks (GNNs) to effectively capture essential features of MILP instances [8, 18]. Using this representation, we recast the original task as a graph generation problem. However, generating such complex graphs from scratch can be computationally expensive and may destroy the intrinsic combinatorial structures of the problems [19]. To address this issue, we propose a masked variational autoencoder (VAE) paradigm inspired by masked language models (MLM) [20, 21] and VAE theories [22–24]. The proposed paradigm iteratively corrupts and replaces parts of the original graphs using sampled latent vectors. This approach allows for controlling the degree to which we change the original instances, thus balancing the novelty and the preservation of structures and hardness of the generated instances. To implement the complicated generation steps, we design a decoder consisting of four modules that work cooperatively to determine multiple components of new instances, involving both structure and numerical prediction tasks simultaneously.

We then design a suite of benchmarks to evaluate the quality of generated MILP instances. First, we measure the structural distributional similarity between the generated samples and the input training instances using multiple structural statistics. Second, we solve the instances using the advanced solver Gurobi [12], and we report the solving time and the numbers of branching nodes of the instances, which directly indicate their computational hardness [19, 25]. Our experiments demonstrate that G2MILP is the very first method capable of generating instances that closely resemble the training sets in terms of both structures and computational hardness. Furthermore, we show that G2MILP is able to adjust the trade-off between the novelty and the preservation of structures and hardness of the generated instances. Third, we conduct a downstream task, the optimal value prediction task, to demonstrate the potential of generated instances in enhancing MILP solvers. The results show that using the generated instances to enrich the training sets reduces the prediction error by over 20% on several datasets. The deliverables are released at https://miralab-ustc.github.io/L2O-G2MILP.

## 2   Related Work

**Machine Learning for MILP**    Machine learning (ML) techniques, due to its capability of capturing rich features from data, has shown impressive potential in addressing combinatorial optimization (CO) problems [26–28], especially MILP problems [5]. Some works apply ML models to directly predict the solutions for MILPs [29–31]. Others attempt to incorporate ML models into heuristic components in modern solvers [7, 9, 32, 33]. Gasse et al. [8] proposed to represent MILP instances as bipartite graphs, and use graph neural networks (GNNs) to capture features for branching decisions. Our

proposed generative framework can produce novel instances to enrich the datasets, which promises to enhance the existing ML methods that require large amounts of i.i.d. training data.

**MILP Instance Generation** Many previous works have made efforts to generate synthetic MILP instances for developing and testing solvers. Existing methods fall into two categories. The first category focuses on using mathematical formulations to generate instances for specific combinatorial optimization problems such as TSP [14], set covering [15], and mixed-integer knapsack [34]. The second category aims to generate general MILP instances. Bowly [16] proposed a framework to generate feasible and bounded MILP instances by sampling from an encoding space that controls a few specific statistics, e.g., density, node degrees, and coefficient mean. However, the aforementioned methods either rely heavily on expert-designed formulations or struggle to capture the rich features of real-world instances. G2MILP tackles these two issues simultaneously by employing deep learning techniques to actively generate instances that resemble real-world problems, and it provides a versatile solution to the data limitation challenge. In [35], we further extend G2MILP to learn to generate challenging MILP instance.

**Deep Graph Generation** A plethora of literature has investigated deep learning models for graph generation [36], including auto-regressive methods [37], varational autoencoders (VAEs) [23], and generative diffusion models [38]. These models have been widely used in various fields [39] such as molecule design [21, 40, 41] and social network generation [42, 43]. G2SAT [17], the first deep learning method for SAT instance generation, has received much research attention [19, 44]. Nevertheless, it is non-trivial to adopt G2SAT to MILP instance generation (see Appendix C.1), as G2SAT does not consider the high-precision numerical prediction, which is one of the fundamental challenges in MILP instance generation. In this paper, we propose G2MILP—the first deep generative framework designed for general MILP instances—and we hope to open up a new research direction for the research community.

## 3 Methodology

In this section, we present our G2MILP framework. First, in Section 3.1, we describe the approach to representing MILP instances as bipartite graphs. Then, in Section 3.2, we derive the masked variational autoencoder (VAE) generative paradigm. In Section 3.3, we provide details on the implementation of the model framework. Finally, in Section 3.4, we explain the training and inference processes. The model overview is in Figure 1. More implementation details can be found in Appendix A. The code is released at https://github.com/MIRALab-USTC/L2O-G2MILP.

### 3.1 Data Representation

A mixed-linear programming (MILP) problem takes the form of:

$$\min_{\boldsymbol{x} \in \mathbb{R}^n} \boldsymbol{c}^\top \boldsymbol{x}, \quad \text{s.t. } \boldsymbol{A}\boldsymbol{x} \leq \boldsymbol{b}, \, \boldsymbol{l} \leq \boldsymbol{x} \leq \boldsymbol{u}, \, x_j \in \mathbb{Z}, \, \forall j \in \mathcal{I}, \tag{1}$$

where $\boldsymbol{c} \in \mathbb{R}^n, \boldsymbol{A} \in \mathbb{R}^{m \times n}, \boldsymbol{b} \in \mathbb{R}^m, \boldsymbol{l} \in (\mathbb{R} \cup \{-\infty\})^n, \boldsymbol{u} \in (\mathbb{R} \cup \{+\infty\})^n$, and the index set $\mathcal{I} \subset \{1, 2, \cdots, n\}$ includes those indices $j$ where $x_j$ is constrained to be an integer.

To represent each MILP instance, we construct a weighted bipartite graph $\mathcal{G} = (\mathcal{V} \cup \mathcal{W}, \mathcal{E})$ as follows [18, 29].

- The constraint vertex set $\mathcal{V} = \{v_1, \cdots, v_m\}$, where each $v_i$ corresponds to the $i^{\text{th}}$ constraint in Equation 1. The vertex feature $\boldsymbol{v}_i$ of $v_i$ is described by the bias term, i.e., $\boldsymbol{v}_i = (b_i)$.

- The variable vertex set $\mathcal{W} = \{w_1, \cdots, w_n\}$, where each $w_j$ corresponds to the $j^{\text{th}}$ variable in Equation 1. The vertex feature $\boldsymbol{w}_j$ of $w_j$ is a 9-dimensional vector that contains information of the objective coefficient $c_j$, the variable type, and the bounds $l_j, u_j$.

- The edge set $\mathcal{E} = \{e_{ij}\}$, where an edge $e_{ij}$ connects a constraint vertex $v_i \in \mathcal{V}$ and a variable vertex $w_j \in \mathcal{W}$. The edge feature $\boldsymbol{e}_{ij}$ is described by the coefficient, i.e., $\boldsymbol{e}_{ij} = (a_{ij})$, and there is no edge between $v_i$ and $w_j$ if $a_{ij} = 0$.

As described above, each MILP instance is represented as a weighted bipartite graph, equipped with a tuple of feature matrices $(\boldsymbol{V}, \boldsymbol{W}, \boldsymbol{E})$, where $\boldsymbol{V}, \boldsymbol{W}, \boldsymbol{E}$ denote stacks of vertex features $\boldsymbol{v}_i, \boldsymbol{w}_j$ and

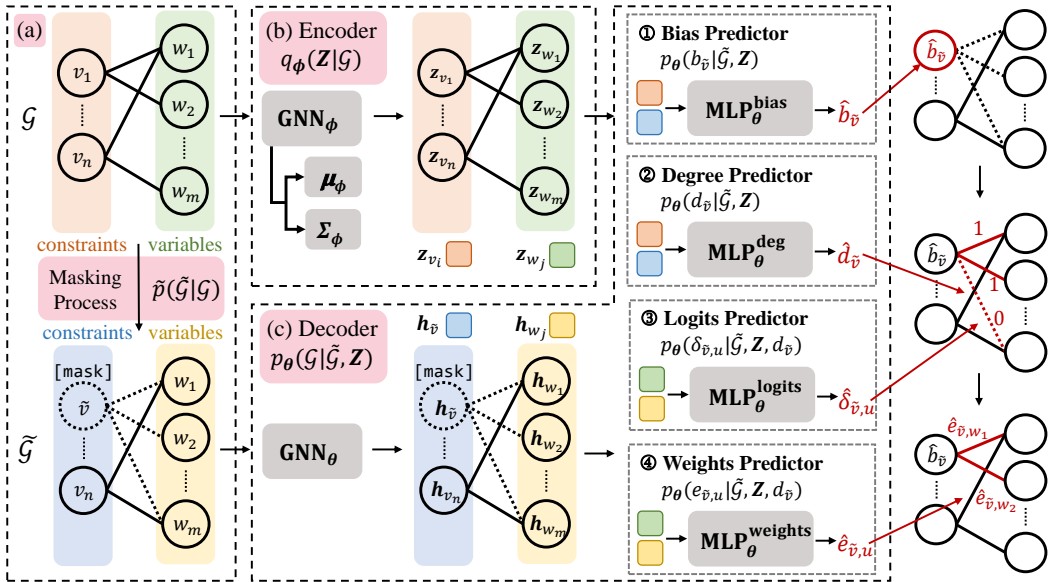

Figure 1: Overview of G2MILP. **(a) Masking Process** $\tilde{p}(\tilde{\mathcal{G}}|\mathcal{G})$**.** Given a MILP instance, which is represented as a bipartite graph $\mathcal{G}$, we randomly label a constraint vertex $\tilde{v}$ as [mask] to obtain the masked graph $\tilde{\mathcal{G}}$. **(b) Encoder** $q_{\boldsymbol{\phi}}(\mathbf{Z}|\mathcal{G})$**.** The encoder is GNN$_{\boldsymbol{\phi}}$ followed by two networks, $\boldsymbol{\mu}_{\boldsymbol{\phi}}$ and $\boldsymbol{\Sigma}_{\boldsymbol{\phi}}$, for resampling. During training, we use the encoder to obtain the latent vectors $\boldsymbol{z}_{v_i}$ and $\boldsymbol{z}_{w_j}$ for all vertices. **(c) Decoder** $p_{\boldsymbol{\theta}}(\mathcal{G}|\tilde{\mathcal{G}}, \mathbf{Z})$**.** We use GNN$_{\boldsymbol{\phi}}$ to obtain the node features $\boldsymbol{h}_{\tilde{v}}$ and $\boldsymbol{h}_{w_j}$. Then four modules work cooperatively to reconstruct the original graph $\mathcal{G}$ based on the node features and the latent vectors. They sequentially determine ① the bias terms, ② the degrees, ③ the logits, and ④ the weights. During inference, the model is decoder-only, and we draw the latent vectors from a standard Guassian distribution to introduce randomness. We repeat the above mask-and-generate process several times so as to produce new instances.

edge features $\boldsymbol{e}_{ij}$, respectively. Such a representation contains all information of the original MILP instance [18]. We use the off-the-shelf observation function provided by Ecole [45] to build the bipartite graphs from MILP instances. We then apply a graph neural network (GNN) to obtain the node representations $\boldsymbol{h}_{v_i}^{\mathcal{G}}$ and $\boldsymbol{h}_{w_j}^{\mathcal{G}}$, also denoted as $\boldsymbol{h}_{v_i}$ and $\boldsymbol{h}_{w_j}$ for simplicity. More details on the data representation can be found in Appendix A.1.

### 3.2 Masked VAE Paradigm

We then introduce our proposed masked VAE paradigm. For the ease of understanding, we provide an intuitive explanation here, and delay the mathematical derivation to Appendix A.2.

Given a graph $\mathcal{G}$ drawn from a dataset $\mathcal{D}$, we corrupt it through a masking process, denoted by $\tilde{\mathcal{G}} \sim \tilde{p}(\tilde{\mathcal{G}}|\mathcal{G})$. We aim to build a parameterized generator $p_{\boldsymbol{\theta}}(\hat{\mathcal{G}}|\tilde{\mathcal{G}})$ that can generate new instances $\hat{\mathcal{G}}$ from the corrupted graph $\tilde{\mathcal{G}}$. We train the generator by maximizing the log-likelihood $\log p_{\boldsymbol{\theta}}(\mathcal{G}|\tilde{\mathcal{G}}) = \log p_{\boldsymbol{\theta}}(\hat{\mathcal{G}} = \mathcal{G}|\tilde{\mathcal{G}})$ of reconstructing $\mathcal{G}$ given $\tilde{\mathcal{G}}$. Therefore, the optimization objective is:

$$\arg \max_{\boldsymbol{\theta}} \mathbb{E}_{\mathcal{G} \sim \mathcal{D}} \mathbb{E}_{\tilde{\mathcal{G}} \sim \tilde{p}(\tilde{\mathcal{G}}|\mathcal{G})} \log p_{\boldsymbol{\theta}}(\mathcal{G}|\tilde{\mathcal{G}}). \tag{2}$$

To model the randomnesss in the generation process and produce diverse instances, we follow the standard VAE framework [22, 23] to introduce a latent variable $\mathbf{Z} = (\boldsymbol{z}_{v_1}, \cdots, \boldsymbol{z}_{v_m}, \boldsymbol{z}_{w_1}, \cdots, \boldsymbol{z}_{w_n})$, which contains the latent vectors for all vertices. During training, the latent vectors are sampled from a posterior distribution given by a parameterized encoder $q_{\boldsymbol{\phi}}$, while during inference, they are independently sampled from a prior distribution such as a standard Gaussian distribution. The decoder $p_{\boldsymbol{\theta}}$ in the masked VAE framework generates new instances from the masked graph $\tilde{\mathcal{G}}$ together with the sampled latent variable $\mathbf{Z}$.

The evidence lower bound (ELBO), also known as the variational lower bound, is a lower bound estimator of the log-likelihood, and is what we actually optimize during training, because it is more tractable. We can derive the ELBO as:

$$\log p_{\boldsymbol{\theta}}(\mathcal{G}|\tilde{\mathcal{G}}) \geq \mathbb{E}_{\mathbf{Z} \sim q_{\boldsymbol{\phi}}(\mathbf{Z}|\mathcal{G})} \left[\log p_{\boldsymbol{\theta}}(\mathcal{G}|\tilde{\mathcal{G}}, \mathbf{Z})\right] - D_{\mathrm{KL}}\left[q_{\boldsymbol{\phi}}(\mathbf{Z}|\mathcal{G}) \| p_{\boldsymbol{\theta}}(\mathbf{Z})\right], \tag{3}$$

where $p_{\boldsymbol{\theta}}(\mathbf{Z})$ is the prior distribution of $\mathbf{Z}$ and is usually taken as the standard Gaussian $\mathcal{N}(\mathbf{0}, \boldsymbol{I})$, and $D_{\mathrm{KL}}[\cdot \| \cdot]$ denotes the KL divergence. Therefore, we formulate the loss function as:

$$\mathcal{L} = \mathbb{E}_{\mathcal{G} \sim \mathcal{D}} \mathbb{E}_{\tilde{\mathcal{G}} \sim \tilde{p}(\tilde{\mathcal{G}}|\mathcal{G})} \left[ \underbrace{\mathbb{E}_{\mathbf{Z} \sim q_{\boldsymbol{\phi}}(\mathbf{Z}|\mathcal{G})} \left[ -\log p_{\boldsymbol{\theta}}(\mathcal{G}|\tilde{\mathcal{G}}, \mathbf{Z}) \right]}_{\mathcal{L}_{\mathrm{rec}}} + \beta \cdot \underbrace{D_{\mathrm{KL}}\left[q_{\boldsymbol{\phi}}(\mathbf{Z}|\mathcal{G}) \| \mathcal{N}(\mathbf{0}, \boldsymbol{I})\right]}_{\mathcal{L}_{\mathrm{prior}}} \right]. \tag{4}$$

In the formula: (1) the first term $\mathcal{L}_{\mathrm{rec}}$ is the reconstruction loss, which urges the decoder to rebuild the input data according to the masked data and the latent variables. (2) The second term $\mathcal{L}_{\mathrm{prior}}$ is used to regularize the posterior distribution in the latent space to approach a standard Gaussian distribution, so that we can sample $\mathbf{Z}$ from the distribution when inference. (3) $\beta$ is a hyperparameter to control the weight of regularization, which is critical in training a VAE model [46].

### 3.3 Model Implementation

To implement Equation 4, we need to instantiate the masking process $\tilde{p}(\tilde{\mathcal{G}}|\mathcal{G})$, the encoder $q_{\boldsymbol{\phi}}(\mathbf{Z}|\mathcal{G})$, and the decoder $p_{\boldsymbol{\theta}}(\mathcal{G}|\tilde{\mathcal{G}}, \mathbf{Z})$, respectively.

**Masking Process** For simplicity, we uniformly sample a constraint vertex $\tilde{v} \sim \mathcal{U}(\mathcal{V})$ and mask it and its adjacent edges, while keeping the variable vertices unchanged. Specifically, we label the vertex $\tilde{v}$ with a special [mask] token, and add virtual edges that link $\tilde{v}$ with each variable vertex. The vertex $\tilde{v}$ and the virtual edges are assigned special embeddings to distinguish them from the others. We further discuss on the masking scheme in Appendix C.2.

**Encoder** The encoder $q_{\boldsymbol{\phi}}(\mathbf{Z}|\mathcal{G})$ is implemented as:

$$q_{\boldsymbol{\phi}}(\mathbf{Z}|\mathcal{G}) = \prod_{u \in \mathcal{V} \cup \mathcal{W}} q_{\boldsymbol{\phi}}(\boldsymbol{z}_u|\mathcal{G}), \quad q_{\boldsymbol{\phi}}(\boldsymbol{z}_u|\mathcal{G}) = \mathcal{N}(\boldsymbol{\mu}_{\boldsymbol{\phi}}(\boldsymbol{h}_u^{\mathcal{G}}), \exp \boldsymbol{\Sigma}_{\boldsymbol{\phi}}(\boldsymbol{h}_u^{\mathcal{G}})), \tag{5}$$

where $\boldsymbol{h}_u^{\mathcal{G}}$ is the node representation of $u$ obtained by a $\mathrm{GNN}_{\boldsymbol{\phi}}$, $\mathcal{N}$ denotes the Gaussian distribution, and $\boldsymbol{\mu}$ and $\boldsymbol{\Sigma}$ output the mean and the log variance, respectively.

**Decoder** The decode $p_{\boldsymbol{\theta}}$ aims to reconstruct $\mathcal{G}$ during training. We apply a $\mathrm{GNN}_{\boldsymbol{\theta}}$ to obtain the node representations $\boldsymbol{h}_u^{\tilde{\mathcal{G}}}$, denoted as $\boldsymbol{h}_u$ for simplicity. To rebuild the masked constraint vertex $\tilde{v}$, the decoder sequentially determines: ① the bias $b_{\tilde{v}}$ (i.e., the right-hand side of the constraint), ② the degree $d_{\tilde{v}}$ of $\tilde{v}$ (i.e., the number of variables involved in the constraint), ③ the logits $\delta_{\tilde{v},u}$ for all variable vertices $u$ to indicate whether they are connected with $\tilde{v}$ (i.e., whether the variables are in the constraint), and ④ the weights $e_{\tilde{v},u}$ of the edges (i.e., the coefficients of the variables in the constraint). The decoder is then formulated as:

$$p_{\boldsymbol{\theta}}(\mathcal{G}|\tilde{\mathcal{G}}, \mathbf{Z}) = p_{\boldsymbol{\theta}}(b_{\tilde{v}}|\tilde{\mathcal{G}}, \mathbf{Z}) \cdot p_{\boldsymbol{\theta}}(d_{\tilde{v}}|\tilde{\mathcal{G}}, \mathbf{Z}) \cdot \prod_{u \in \mathcal{W}} p_{\boldsymbol{\theta}}(\delta_{\tilde{v},u}|\tilde{\mathcal{G}}, \mathbf{Z}, d_{\tilde{v}}) \cdot \prod_{u \in \mathcal{W}: \delta_{\tilde{v},u}=1} p_{\boldsymbol{\theta}}(e_{\tilde{v},u}|\tilde{\mathcal{G}}, \mathbf{Z}). \tag{6}$$

Therefore, we implement the decoder as four cooperative modules: ① Bias Predictor, ② Degree Predictor, ③ Logits Predictor, and ④ Weights Predictor.

① **Bias Predictor** For effective prediction, we incorporate the prior of simple statistics of the dataset—the minimum $\underline{b}$ and the maximum $\overline{b}$ of the bias terms that occur in the dataset—into the predictor. Specifically, we normalize the bias $b_{\tilde{v}}$ to $[0, 1]$ via $b_{\tilde{v}}^* = (b_{\tilde{v}} - \underline{b})/(\overline{b} - \underline{b})$. To predict $b_{\tilde{v}}^*$, we use one MLP that takes the node representation $\boldsymbol{h}_{\tilde{v}}$ and the latent vector $\boldsymbol{z}_{\tilde{v}}$ of $\tilde{v}$ as inputs:

$$\hat{b}_{\tilde{v}}^* = \sigma\left(\mathrm{MLP}_{\boldsymbol{\theta}}^{\mathrm{bias}}([\boldsymbol{h}_{\tilde{v}}, \boldsymbol{z}_{\tilde{v}}])\right), \tag{7}$$

where $\sigma(\cdot)$ is the sigmoid function used to restrict the outputs. We use the mean squared error (MSE) loss to train the predictor. At inference time, we apply the inverse transformation to obtain the predicted bias values: $\hat{b}_{\tilde{v}} = \underline{b} + (\overline{b} - \underline{b}) \cdot \hat{b}_{\tilde{v}}^*$.[1]

---

[1]Notation-wise, we use $\hat{x}$ to denote the predicted variable in $\hat{\mathcal{G}}$ that corresponds to $x$ in $\mathcal{G}$.

② **Degree Predictor** We find that the constraint degrees are crucial to the graph structures and significantly affect the combinatorial properties. Therefore, we use the Degree Predictor to determine coarse-grained degree structure, and then use the Logits Predictor to determine the fine-grained connection details. Similarly to Bias Predictor, we normalize the degree $d_{\tilde{v}}$ to $d_{\tilde{v}}^* = (d_{\tilde{v}} - \underline{d})/(\overline{d} - \underline{d})$, where $\underline{d}$ and $\overline{d}$ are the minimum and maximum degrees in the dataset, respectively. We use one MLP to predict $d_{\tilde{v}}^*$:

$$\hat{d}_{\tilde{v}}^* = \sigma\left(\text{MLP}_{\boldsymbol{\theta}}^{\text{deg}}([\boldsymbol{h}_{\tilde{v}}, \boldsymbol{z}_{\tilde{v}}])\right). \tag{8}$$

We use MSE loss for training, and round the predicted degree to the nearest integer $\hat{d}_{\tilde{v}}$ for inference.

③ **Logits Predictor** To predict the logits $\delta_{\tilde{v},u}$ indicating whether a variable vertex $u \in \mathcal{W}$ is connected with $\tilde{v}$, we use one MLP that takes the representation $\boldsymbol{h}_u$ and the latent vector $\boldsymbol{z}_u$ of $u$ as inputs:

$$\hat{\delta}_{\tilde{v},u}' = \sigma\left(\text{MLP}_{\boldsymbol{\theta}}^{\text{logits}}([\boldsymbol{h}_u, \boldsymbol{z}_u])\right). \tag{9}$$

We use binary cross-entropy (BCE) loss to train the logistical regression module. As positive samples (i.e., variables connected with a constraint) are often scarce, we use one negative sample for each positive sample during training. The loss function is:

$$\mathcal{L}_{\text{logits}} = -\mathbb{E}_{(\tilde{v},u)\sim p_{\text{pos}}}\left[\log\left(\hat{\delta}_{\tilde{v},u}'\right)\right] - \mathbb{E}_{(\tilde{v},u)\sim p_{\text{neg}}}\left[\log\left(1 - \hat{\delta}_{\tilde{v},u}'\right)\right], \tag{10}$$

where $p_{\text{pos}}$ and $p_{\text{neg}}$ denote the distributions over positive and negative samples, respectively. At inference time, we connect $\hat{d}_{\tilde{v}}$ variable vertices with the top logits to $\tilde{v}$., i.e.,

$$\hat{\delta}_{\tilde{v},u} = \begin{cases} 1, u \in \arg\text{TopK}(\{\hat{\delta}_{\tilde{v},u}'|u \in \mathcal{W}\}, \hat{d}_{\tilde{v}}), \\ 0, \text{otherwise}. \end{cases} \tag{11}$$

④ **Weights Predictor** Finally, we use one MLP to predict the normalized weights $e_{\tilde{v},u}^*$ for nodes $u$ that are connected with $\tilde{v}$:

$$\hat{e}_{\tilde{v},u}^* = \sigma\left(\text{MLP}_{\boldsymbol{\theta}}^{\text{weights}}([\boldsymbol{h}_u, \boldsymbol{z}_u])\right). \tag{12}$$

The training and inference procedures are similar to those of Bias Predictor.

### 3.4 Training and Inference

During training, we use the original graph $\mathcal{G}$ to provide supervision signals for the decoder, guiding it to reconstruct $\mathcal{G}$ from the masked $\tilde{\mathcal{G}}$ and the encoded $\mathbf{Z}$. As described above, the decoder involves four modules, each optimized by a prediction task. The first term in Equation 4, i.e., the reconstruction loss, is written as

$$\mathcal{L}_{\text{rec}} = \mathbb{E}_{\mathcal{G}\sim\mathcal{D},\tilde{\mathcal{G}}\sim\tilde{p}(\tilde{\mathcal{G}}|\mathcal{G})}\left[\sum_{i=1}^{4} \alpha_i \cdot \mathcal{L}_i(\boldsymbol{\theta}, \boldsymbol{\phi}|\mathcal{G}, \tilde{\mathcal{G}})\right], \tag{13}$$

where $\mathcal{L}_i(\boldsymbol{\theta}, \boldsymbol{\phi}|\mathcal{G}, \tilde{\mathcal{G}})$ $(i = 1, 2, 3, 4)$ are loss functions for the four prediction tasks, respectively, and $\alpha_i$ are hyperparameters.

During inference, we discard the encoder and sample $\mathbf{Z}$ from a standard Gaussian distribution, which introduces randomness to enable the model to generate novel instances. We iteratively mask one constraint vertex in the bipartite graph and replace it with a generated one. We define a hyperparameter $\eta$ to adjust the ratio of iterations to the number of constraints, i.e., $N_{\text{iters}} = \eta \cdot |\mathcal{V}|$. Naturally, a larger value of $\eta$ results in instances that are more novel, while a smaller value of $\eta$ yields instances that exhibit better similarity to the training set. For further details of training and inference procedures, please refer to Appendix A.3.

## 4 Experiments

### 4.1 Setup

We conduct extensive experiments to demonstrate the effectiveness of our model. More experimental details can be found in Appendix B. Additional results are in Appendix C.

**Benchmarking**    To evaluate the quality of the generated MILP instances, we design three benchmarks so as to answer the following research questions. (1) How well can the generated instances preserve the graph structures of the training set? (2) How closely do the generated instances resemble the computational hardness of real-world instances? (3) How effectively do they facilitate downstream tasks to improve solver performance?

**I. Structural Distributional Similarity**    We consider 11 classical statistics to represent features of the instances [17, 47], including coefficient density, node degrees, graph clustering, graph modularity, etc. In line with a widely used graph generation benchmark [48], we compute the Jensen-Shannon (JS) divergence [49] for each statistic to measure the distributional similarity between the generated instances and the training set. We then standardize the metrics into similarity scores that range from 0 to 1. The computing details can be found in Appendix B.3.

**II. Computational Hardness**    The computational hardness is another critical metric to assess the quality of the generated instances. We draw an analogy from the SAT generation community, where though many progresses achieved, it is widely acknowledged that the generated SAT instances differs significantly from real-world ones in the computational hardness [25], and this issue remains inadequately addressed. In our work, we make efforts to mitigate this problem, even in the context of MILP generation, a more challenging task. To this end, we leverage the state-of-the-art solver Gurobi [12] to solve the instances, and we report the solving time and the numbers of branching nodes during the solving process, which can directly reflect the computational hardness of instances [19].

**III. Downstream Task**    We consider two downstream tasks to examine the the potential benefits of the generated instances in practical applications. We employ G2MILP to generate new instances and augment the original datasets, and then evaluate whether the enriched datasets can improve the performance of the downstream tasks. The considered tasks include predicting the optimal values of the MILP problem, as discussed in Chen et al. [18], and applying a predict-and-search framework for solving MILPs, as proposed by Han et al. [31].

**Datasets**    We consider four different datasets of various sizes. (1) *Large datasets.* We evaluate the model's capability of learning data distributions using two well-known synthetic MILP datasets: Maximum Independent Set (MIS) [50] and Set Covering [15]. We follow previous works [8, 9] to artificially generate 1000 instances for each of them. (2) *Medium dataset.* Mixed-integer Knapsack (MIK) is a widely used dataset [34], which consists of 80 training instances and 10 test instances. We use this dataset to evaluate the model's performance both on the distribution learning benchmarks and the downstream task. (3) *Small dataset.* We construct a small subset of MIPLIB 2017 [10] by collecting a group of problems called Nurse Scheduling problems. This dataset comes from real-world scenarios and consists of only 4 instances, 2 for training and 2 for test, respectively. Since the statistics are meaningless for such an extremely small dataset, we use it only to demonstrate the effectiveness of generated instances in facilitating downstream tasks.

**Baselines**    G2MILP is the first deep learning generative framework for MILP isntances, and thus, we do not have any learning-based models for comparison purpose. Therefore, we compare G2MILP with a heuristic MILP instance generator, namely Bowly [16]. Bowly can create feasible and bounded MILP instances while controlling some specific statistical features such as coefficient density and coefficient mean. We set all the controllable parameters to match the corresponding statistics of the training set, allowing Bowly to imitate the training set to some extent. We also consider a useful baseline, namely Random, to demonstrate the effectiveness of deep neural networks in G2MILP. Random employs the same generation procedure as G2MILP, but replaces all neural networks in the decoder with random generators. We set the masking ratio $\eta$ for Random and G2MILP to 0.01, 0.05, and 0.1 to show how this hyperparameter helps balance the novelty and similarity.

### 4.2   Quantitative Results

**I. Structural Distributional Similarity**    We present the structural distributional similarity scores between each pair of datasets in Table 1. The results indicate that our designed metric is reasonable in the sense that datasets obtain high scores with themselves and low

Table 1: Structural similarity scores between each pair of datasets. Higher is better.

|          | MIS   | SetCover | MIK   |
|----------|-------|----------|-------|
| MIS      | 0.998 | 0.182    | 0.042 |
| SetCover | -     | 1.000    | 0.128 |
| MIK      | -     | -        | 0.997 |

Table 3: Average solving time (s) of instances solved by Gurobi (mean ± std). $\eta$ is the masking ratio. Numbers in the parentheses are relative errors with respect to the training sets (lower is better).

|  |  | MIS | SetCover | MIK |
|---|---|---|---|---|
| Training Set | | 0.349 ± 0.05 | 2.344± 0.13 | 0.198± 0.04 |
| Bowly | | 0.007± 0.00 (97.9%) | 0.048± 0.00 (97.9%) | 0.001± 0.00 (99.8%) |
| $\eta = 0.01$ | Random | 0.311± 0.05 (10.8%) | 2.044± 0.19 (12.8%) | 0.008± 0.00 (96.1%) |
| | G2MILP | **0.354± 0.06 (1.5%)** | **2.360± 0.18 (0.8%)** | **0.169± 0.07 (14.7%)** |
| $\eta = 0.05$ | Random | 0.569± 0.09 (63.0%) | 2.010± 0.11 (14.3%) | 0.004± 0.00 (97.9%) |
| | G2MILP | **0.292± 0.07 (16.3%)** | **2.533± 0.15 ( 8.1%)** | **0.129± 0.05 (35.1%)** |
| $\eta = 0.1$ | Random | 2.367± 0.35 (578.2%) | 1.988± 0.17 (15.2%) | 0.005± 0.00 (97.6%) |
| | G2MILP | **0.214± 0.05 (38.7%)** | **2.108± 0.21 (10.0%)** | **0.072± 0.02 (63.9%)** |

Table 4: Average numbers of branching nodes of instances solved by Gurobi. $\eta$ is the masking ratio. Numbers in the parentheses are relative errors with respect to the training sets (lower is better).

|  |  | MIS | SetCover | MIK |
|---|---|---|---|---|
| Training Set | | 16.09 | 838.56 | 175.35 |
| Bowly | | 0.00 (100.0%) | 0.00 (100.0%) | 0.00 (100.0%) |
| $\eta = 0.01$ | Random | 20.60 (28.1%) | **838.51 (0.0%)** | 0.81 (99.5%) |
| | G2MILP | **15.03 (6.6%)** | 876.09 (4.4%) | **262.25 (14.7%)** |
| $\eta = 0.05$ | Random | 76.22 (373.7%) | 765.30 (8.7%) | 0.00 (100%) |
| | G2MILP | **10.58 (34.2%)** | **874.46 (4.3%)** | **235.35 (34.2%)** |
| $\eta = 0.1$ | Random | 484.47 (2911.2%) | 731.14 (12.8%) | 0.00 (100%) |
| | G2MILP | **4.61 (71.3%)** | **876.92 (4.6%)** | **140.06 (20.1%)** |

scores with different domains. Table 2 shows the similarity scores between generated instances and the corresponding training sets. We generate 1000 instances for each dataset to compute the similarity scores. The results suggest that G2MILP closely fits the data distribution, while Bowly, which relies on heuristic rules to control the statistical features, falls short of our expectations. Furthermore, we observe that G2MILP outperforms Random, indicating that deep learning contributes to the model's performance. As expected, a higher masking ratio $\eta$ results in generating more novel instances but reduces their similarity to the training sets.

**II. Computational Hardness**  We report the average solving time and numbers of branching nodes in Table 3 and Table 4, respectively. The results indicate that instances generated by Bowly are relatively easy, and the hardness of those generated by Random is inconclusive. In contrast, G2MILP is capable of preserving the computational hardness of the original training sets. Notably, even without imposing rules to guarantee the feasibility and boundedness of generated instances, G2MILP automatically learns from the data and produces feasible and bounded instances.

Table 2: Structural distributional similarity scores between the generated instances with the training datasets. Higher is better. $\eta$ is the masking ratio. We do not report the results of Bowly on MIK because Ecole [45] and SCIP [51] fail to read the generated instances due to large numerical values.

|  |  | MIS | SetCover | MIK |
|---|---|---|---|---|
| Bowly | | 0.184 | 0.197 | - |
| $\eta = 0.01$ | Random | 0.651 | 0.735 | 0.969 |
| | G2MILP | **0.997** | **0.835** | **0.991** |
| $\eta = 0.05$ | Random | 0.580 | 0.613 | 0.840 |
| | G2MILP | **0.940** | **0.782** | **0.953** |
| $\eta = 0.1$ | Random | 0.512 | 0.556 | 0.700 |
| | G2MILP | **0.895** | **0.782** | **0.918** |

**III. Downstream Task**  First, we follow Chen et al. [18] to construct a GNN model for predicting the optimal values of MILP problems. We train a predictive GNN model on the training set. After that, we employ 20 generated instances to augment the training data, and then train another predictive

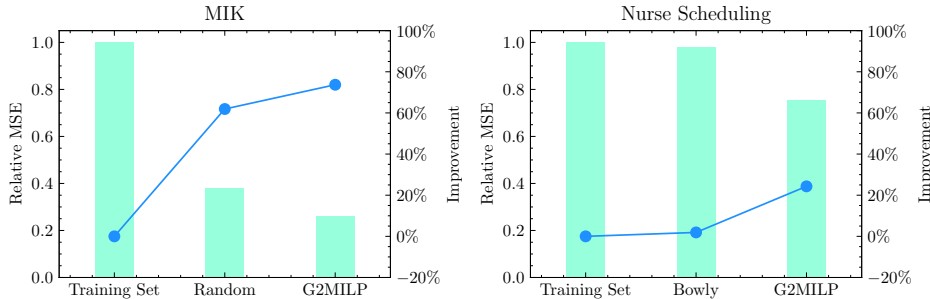

Figure 2: Results of the optimal value prediction task. Bars indicate the relative MSE to the model trained on the original training sets, and lines represent the relative performance improvement.

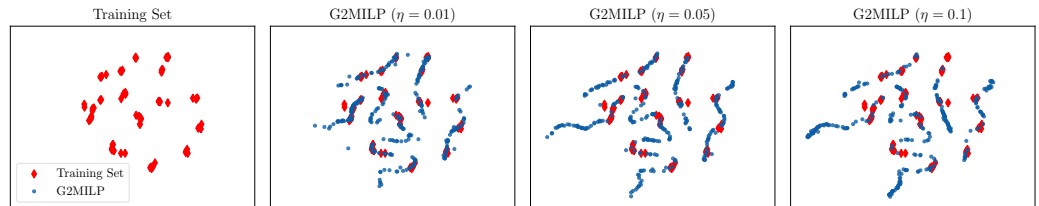

Figure 3: The t-SNE visualization of MILP instance representations for MIK. Each point represents an instance. Red points are from the training set and blue points are instances generated by G2MILP.

model using the enriched dataset. We use the prediction mean squared error (MSE) to assess the resulting models, and we present the MSE relative to the default model trained on the original training sets in Figure 2. For the MIK dataset, instances generated by Bowly introduce numerical issues so that Ecole and SCIP fail to read them. For the Nurse Scheduling dataset, Random fails to generate feasible instances. Notably, G2MILP is the only method that demonstrates performance improvement on both datasets, reducing the MSE by 73.7% and 24.3%, respectively. The detailed results are in Table 8 in Appendix B.4. Then, we conduct experiments on the neural solver, i.e., the predict-and-search framework proposed by Han et al. [31], which employs a model to predict a solution and then uses solvers to search for the optimal solutions in a trust region. The results are in Table 9 in Appendix B.4.

## 4.3 Analysis

**Masking Process**  We conduct extensive comparative experiments on different implementations of the masking process. First, we implement different versions of G2MILP, which enable us to mask and modify either constraints, variables, or both. Second, we investigate different orders of masking constraints, including uniformly sampling and sampling according to the vertex indices. Third, we analyze the effect of the masking ratio $\eta$ on similarity scores and downstream task performance improvements. The experimental results are in Appendix C.2.

**Size of Dataset**  We conduct experiments on different sizes of the original datasets and different ratio of generated instances to original ones on MIS. Results are in Table 15 in Appendix C.4. The results show that G2MILP yields performance improvements across datasets of varying sizes.

**Visualization**  We visualize the instance representations for MIK in Figure 3. Specifically, we use the G2MILP encoder to obtain the instance representations, and then apply t-SNE dimensionality reduction [52] for visualization. We observe that the generated instances, while closely resembling the training set, contribute to a broader and more continuous exploration of the problem space, thereby enhancing model robustness and generalization. Additionally, by increasing the masking ratio $\eta$, we can effectively explore a wider problem space beyond the confines of the training sets. For comparison with the baseline, we present the visualization of instances generated by Random in Figure 5 in Appendix C.5.

# 5   Limitations, Future Avenues, and Conclusions

**Limitations**   In this paper, we develop G2MILP by iteratively corrupting and replacing the constraints vertices. We also investigate different implementations of the masking process. However, more versatile masking schemes should be explored. Moreover, employing more sophisticated designs would enable us to control critical properties such as feasibility of the instances. We intend to develop a more versatile and powerful generator in our future work.

**Future Avenues**   We open up new avenues for research on DL-based MILP instance generative models. In addition to producing new instances to enrich the datasets, this research has many other promising technical implications [35]. (1) Such a generator will assist researchers to gain insights into different data domains and the explored space of MILP instances. (2) Based on a generative model, we can establish an adversarial framework, where the generator aims to identify challenging cases for the solver, thus automatically enhancing the solver's ability to handle complex scenarios. (3) Training a generative model involves learning the data distribution and deriving representations through unsupervised learning. Consequently, it is possible to develop a pre-trained model based on a generative model, which can benefit downstream tasks across various domains. We believe that this paper serves as an entrance for the aforementioned routes, and we expect further efforts in this field.

**Conclusions**   In this paper, we propose G2MILP, which to the best of our knowledge is the first deep generative framework for MILP instances. It can learn to generate MILP instances without prior expert-designed formulations, while preserving the structures and computational hardness, simultaneously. Thus the generated instances can enhance MILP solvers under limited data availability. This work opens up new avenues for research on DL-based MILP instance generative models.

## Acknowledgements

The authors would like to thank all the anonymous reviewers for their insightful comments. This work was supported in part by National Key R&D Program of China under contract 2022ZD0119801, National Nature Science Foundations of China grants U19B2026, U19B2044, 61836011, 62021001, and 61836006.

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

# A    Implementation Details

## A.1    Data Representation

As described in the main paper, we represent each MILP instance as a weighted bipartite graph $\mathcal{G} = (\mathcal{V} \cup \mathcal{W}, \mathcal{E})$, where $\mathcal{V}$ represents the constraint vertex set, $\mathcal{W}$ represents the variable vertex set, and $\mathcal{E}$ represents the edge set, respectively. The graph is equipped with a tuple of feature matrices $(\mathbf{V}, \mathbf{W}, \mathbf{E})$, and the description of these features can be found in Table 5.

Table 5: Description of the constraint, variable, and edge features in our bipartite graph representation.

| Tensor | Feature | Description |
|---|---|---|
| **V** | bias | The bias value $b_i$. |
| | type | Variable type (binary, continuous, integer, implicit integer) as a 4-dimensional one-hot encoding. |
| | objective | Objective coefficient $c_j$. |
| **W** | has_lower_bound | Lower bound indicator. |
| | has_upper_bound | Upper bound indicator. |
| | lower_bound | Lower bound value $l_j$. |
| | upper_bound | Upper bound value $u_j$. |
| **E** | coef | Constraint coefficient $a_{ij}$. |

To ensure consistency, we standardize each instance to the form of Equation 1. However, we do not perform data normalization in order to preserve the potential information related to the problem domain in the original formulation. When extracting the bipartite graph, we utilize the readily available observation function provided by Ecole. For additional details on the observation function, readers can consult the following link: `https://doc.ecole.ai/py/en/stable/reference/observations.html#ecole.observation.MilpBipartite`.

## A.2    The Derivation of Masked Variational Auto-Encoder

We consider a variable with a distribution $p(\boldsymbol{x})$. We draw samples from this distribution and apply a masking process to transform each sample $\boldsymbol{x}$ into $\tilde{\boldsymbol{x}}$ through a given probability $\tilde{p}(\tilde{\boldsymbol{x}}|\boldsymbol{x})$. Our objective is to construct a parameterized generator $p_{\boldsymbol{\theta}}(\boldsymbol{x}|\tilde{\boldsymbol{x}})$ to produce new data based on the the masked data $\tilde{\boldsymbol{x}}$. We assume that the generation process involves an unobserved continuous random variable $\boldsymbol{z}$ that is independent of $\tilde{\boldsymbol{x}}$, i.e., $\boldsymbol{z} \perp \tilde{\boldsymbol{x}}$. Consequently, we obtain the following equation:

$$p_{\boldsymbol{\theta}}(\boldsymbol{x}|\tilde{\boldsymbol{x}}) = \frac{p_{\boldsymbol{\theta}}(\boldsymbol{x}|\boldsymbol{z},\tilde{\boldsymbol{x}})p_{\boldsymbol{\theta}}(\boldsymbol{z}|\tilde{\boldsymbol{x}})}{p_{\boldsymbol{\theta}}(\boldsymbol{z}|\boldsymbol{x},\tilde{\boldsymbol{x}})} = \frac{p_{\boldsymbol{\theta}}(\boldsymbol{x}|\boldsymbol{z},\tilde{\boldsymbol{x}})p_{\boldsymbol{\theta}}(\boldsymbol{z})}{p_{\boldsymbol{\theta}}(\boldsymbol{z}|\boldsymbol{x},\tilde{\boldsymbol{x}})}. \tag{14}$$

We introduce a probabilistic encoder $q_{\boldsymbol{\phi}}(\boldsymbol{z}|\boldsymbol{x})$ for approximating the intractable latent variable distribution. We can then derive the follows:

$$
\begin{aligned}
\log p_{\boldsymbol{\theta}}(\boldsymbol{x}|\tilde{\boldsymbol{x}}) =& \mathbb{E}_{\boldsymbol{z} \sim q_{\boldsymbol{\phi}}(\boldsymbol{z}|\boldsymbol{x})}\left[\log p_{\boldsymbol{\theta}}(\boldsymbol{x}|\tilde{\boldsymbol{x}})\right] \\
=& \mathbb{E}_{\boldsymbol{z} \sim q_{\boldsymbol{\phi}}(\boldsymbol{z}|\boldsymbol{x})}\left[\log \frac{p_{\boldsymbol{\theta}}(\boldsymbol{x}|\boldsymbol{z},\tilde{\boldsymbol{x}})p_{\boldsymbol{\theta}}(\boldsymbol{z})}{q_{\boldsymbol{\phi}}(\boldsymbol{z}|\boldsymbol{x})}\frac{q_{\boldsymbol{\phi}}(\boldsymbol{z}|\boldsymbol{x})}{p_{\boldsymbol{\theta}}(\boldsymbol{z}|\boldsymbol{x},\tilde{\boldsymbol{x}})}\right] \\
=& \mathbb{E}_{\boldsymbol{z} \sim q_{\boldsymbol{\phi}}(\boldsymbol{z}|\boldsymbol{x})}\left[\log \frac{p_{\boldsymbol{\theta}}(\boldsymbol{x}|\boldsymbol{z},\tilde{\boldsymbol{x}})p_{\boldsymbol{\theta}}(\boldsymbol{z})}{q_{\boldsymbol{\phi}}(\boldsymbol{z}|\boldsymbol{x})}\right] + \mathbb{E}_{\boldsymbol{z} \sim q_{\boldsymbol{\phi}}(\boldsymbol{z}|\boldsymbol{x})}\left[\log\left(\frac{q_{\boldsymbol{\phi}}(\boldsymbol{z}|\boldsymbol{x})}{p_{\boldsymbol{\theta}}(\boldsymbol{z}|\boldsymbol{x},\tilde{\boldsymbol{x}})}\right)\right] \\
=& -\mathcal{L}(\boldsymbol{\theta},\boldsymbol{\phi}|\boldsymbol{x},\tilde{\boldsymbol{x}}) + D_{\mathrm{KL}}\left[q_{\boldsymbol{\phi}}(\boldsymbol{z}|\boldsymbol{x})\|p_{\boldsymbol{\theta}}(\boldsymbol{z}|\boldsymbol{x},\tilde{\boldsymbol{x}})\right] \\
\geq& -\mathcal{L}(\boldsymbol{\theta},\boldsymbol{\phi}|\boldsymbol{x},\tilde{\boldsymbol{x}}). \tag{15}
\end{aligned}
$$

In the formula, the term $-\mathcal{L}(\boldsymbol{\theta}, \boldsymbol{\phi}|\boldsymbol{x}, \tilde{\boldsymbol{x}})$ is referred to as the evidence lower bound (ELBO), or the variational lower bound. It can be expressed as:

$$
\begin{aligned}
-\mathcal{L}(\boldsymbol{\theta}, \boldsymbol{\phi}|\boldsymbol{x}, \tilde{\boldsymbol{x}}) &= \mathbb{E}_{\boldsymbol{z} \sim q_{\boldsymbol{\phi}}(\boldsymbol{z}|\boldsymbol{x})} \left[ \log \frac{p_{\boldsymbol{\theta}}(\boldsymbol{x}|\boldsymbol{z}, \tilde{\boldsymbol{x}})p_{\boldsymbol{\theta}}(\boldsymbol{z})}{q_{\boldsymbol{\phi}}(\boldsymbol{z}|\boldsymbol{x})} \right] \\
&= \mathbb{E}_{\boldsymbol{z} \sim q_{\boldsymbol{\phi}}(\boldsymbol{z}|\boldsymbol{x})} \left[ \log p_{\boldsymbol{\theta}}(\boldsymbol{x}|\boldsymbol{z}, \tilde{\boldsymbol{x}}) \right] - \mathbb{E}_{\boldsymbol{z} \sim q_{\boldsymbol{\phi}}(\boldsymbol{z}|\boldsymbol{x})} \left[ \log \frac{q_{\boldsymbol{\phi}}(\boldsymbol{z}|\boldsymbol{x})}{p_{\boldsymbol{\theta}}(\boldsymbol{z})} \right] \\
&= \mathbb{E}_{\boldsymbol{z} \sim q_{\boldsymbol{\phi}}(\boldsymbol{z}|\boldsymbol{x})} \left[ \log p_{\boldsymbol{\theta}}(\boldsymbol{x}|\boldsymbol{z}, \tilde{\boldsymbol{x}}) \right] - D_{\mathrm{KL}} \left[ q_{\boldsymbol{\phi}}(\boldsymbol{z}|\boldsymbol{x}) \| p_{\boldsymbol{\theta}}(\boldsymbol{z}) \right].
\end{aligned} \tag{16}
$$

Consequently, the loss function can be formulated as follows:

$$
\mathcal{L}(\boldsymbol{\theta}, \boldsymbol{\phi}) = \mathbb{E}_{\boldsymbol{x} \sim \mathcal{D}} \mathbb{E}_{\tilde{\boldsymbol{x}} \sim \tilde{p}(\tilde{\boldsymbol{x}}|\boldsymbol{x})} \left[ \mathcal{L}(\boldsymbol{\theta}, \boldsymbol{\phi}|\boldsymbol{x}, \tilde{\boldsymbol{x}}) \right], \tag{17}
$$

where

$$
\mathcal{L}(\boldsymbol{\theta}, \boldsymbol{\phi}|\boldsymbol{x}, \tilde{\boldsymbol{x}}) = \underbrace{\mathbb{E}_{\boldsymbol{z} \sim q_{\boldsymbol{\phi}}(\boldsymbol{z}|\boldsymbol{x})} \left[ -\log p_{\boldsymbol{\theta}}(\boldsymbol{x}|\boldsymbol{z}, \tilde{\boldsymbol{x}}) \right]}_{\mathcal{L}_{\mathrm{rec}}} + \underbrace{D_{\mathrm{KL}} \left[ q_{\boldsymbol{\phi}}(\boldsymbol{z}|\boldsymbol{x}) \| p_{\boldsymbol{\theta}}(\boldsymbol{z}) \right]}_{\mathcal{L}_{\mathrm{prior}}}. \tag{18}
$$

In the formula, the first term $\mathcal{L}_{\mathrm{rec}}$ is referred to as the reconstruction loss, as it urges the decoder to reconstruct the input data $\boldsymbol{x}$. The second term $\mathcal{L}_{\mathrm{prior}}$ is referred to as the prior loss, as it regularizes the posterior distribution $q_{\boldsymbol{\phi}}(\boldsymbol{z}|\boldsymbol{x})$ of the latent variable to approximate the prior distribution $p_{\boldsymbol{\theta}}(\boldsymbol{z})$. In practice, the prior distribution $p_{\boldsymbol{\theta}}(\boldsymbol{z})$ is commonly taken as $\mathcal{N}(\boldsymbol{0}, \boldsymbol{I})$, and a hyperparameter is often introduced as the coefficient for the prior loss. Consequently, the loss function can be expressed as:

$$
\mathcal{L}(\boldsymbol{\theta}, \boldsymbol{\phi}) = \mathbb{E}_{\boldsymbol{x} \sim \mathcal{D}} \mathbb{E}_{\tilde{\boldsymbol{x}} \sim \tilde{p}(\tilde{\boldsymbol{x}}|\boldsymbol{x})} \left[ \mathcal{L}(\boldsymbol{\theta}, \boldsymbol{\phi}|\boldsymbol{x}, \tilde{\boldsymbol{x}}) \right], \tag{19}
$$

where

$$
\begin{aligned}
\mathcal{L}(\boldsymbol{\theta}, \boldsymbol{\phi}|\boldsymbol{x}, \tilde{\boldsymbol{x}}) &= \mathcal{L}_{\mathrm{rec}}(\boldsymbol{\theta}, \boldsymbol{\phi}|\boldsymbol{x}, \tilde{\boldsymbol{x}}) + \beta \cdot \mathcal{L}_{\mathrm{prior}}(\boldsymbol{\phi}|\boldsymbol{x}), \\
\mathcal{L}_{\mathrm{rec}}(\boldsymbol{\theta}, \boldsymbol{\phi}|\boldsymbol{x}, \tilde{\boldsymbol{x}}) &= \mathbb{E}_{\boldsymbol{z} \sim q_{\boldsymbol{\phi}}(\boldsymbol{z}|\boldsymbol{x})} \left[ -\log p_{\boldsymbol{\theta}}(\boldsymbol{x}|\boldsymbol{z}, \tilde{\boldsymbol{x}}) \right], \\
\mathcal{L}_{\mathrm{prior}}(\boldsymbol{\phi}|\boldsymbol{x}) &= D_{\mathrm{KL}} \left[ q_{\boldsymbol{\phi}}(\boldsymbol{z}|\boldsymbol{x}) \| \mathcal{N}(\boldsymbol{0}, \boldsymbol{I}) \right].
\end{aligned} \tag{20}
$$

In G2MILP, the loss function is instantiated as:

$$
\mathcal{L}(\boldsymbol{\theta}, \boldsymbol{\phi}) = \mathbb{E}_{\mathcal{G} \sim \mathcal{D}} \mathbb{E}_{\tilde{\mathcal{G}} \sim \tilde{p}(\tilde{\mathcal{G}}|\mathcal{G})} \left[ \mathcal{L}(\boldsymbol{\theta}, \boldsymbol{\phi}|\mathcal{G}, \tilde{\mathcal{G}}) \right], \tag{21}
$$

where

$$
\begin{aligned}
\mathcal{L}(\boldsymbol{\theta}, \boldsymbol{\phi}|\mathcal{G}, \tilde{\mathcal{G}}) &= \mathcal{L}_{\mathrm{rec}}(\boldsymbol{\theta}, \boldsymbol{\phi}|\mathcal{G}, \tilde{\mathcal{G}}) + \beta \cdot \mathcal{L}_{\mathrm{prior}}(\boldsymbol{\phi}|\mathcal{G}), \\
\mathcal{L}_{\mathrm{rec}}(\boldsymbol{\theta}, \boldsymbol{\phi}|\mathcal{G}, \tilde{\mathcal{G}}) &= \mathbb{E}_{\mathbf{Z} \sim q_{\boldsymbol{\phi}}(\mathbf{Z}|\mathcal{G})} \left[ -\log p_{\boldsymbol{\theta}}(\mathcal{G}|\mathbf{Z}, \tilde{\mathcal{G}}) \right], \\
\mathcal{L}_{\mathrm{prior}}(\boldsymbol{\phi}|\mathcal{G}) &= D_{\mathrm{KL}} \left[ q_{\boldsymbol{\phi}}(\mathbf{Z}|\mathcal{G}) \| \mathcal{N}(\boldsymbol{0}, \boldsymbol{I}) \right].
\end{aligned} \tag{22}
$$

### A.3 G2MILP Implementation

#### A.3.1 Encoder

The encoder implements $q_{\boldsymbol{\phi}}(\mathbf{Z}|\mathcal{G})$ in Equation 22. Given a bipartite graph $\mathcal{G} = (\mathcal{V} \cup \mathcal{W}, \mathcal{E})$ equipped with the feature metrices $(\mathbf{V}, \mathbf{W}, \mathbf{E})$, we employ a GNN structure with parameters $\boldsymbol{\phi}$ to extract the representations. Specifically, we utilize MLPs as embedding layers to obtain the initial embeddings $\boldsymbol{h}_{v_i}^{(0)}$, $\boldsymbol{h}_{w_j}^{(0)}$, and $\boldsymbol{h}_{e_{ij}}$, given by:

$$
\boldsymbol{h}_{v_i}^{(0)} = \mathrm{MLP}_{\boldsymbol{\phi}}(\boldsymbol{v}_i), \quad \boldsymbol{h}_{w_j}^{(0)} = \mathrm{MLP}_{\boldsymbol{\phi}}(\boldsymbol{w}_j), \quad \boldsymbol{h}_{e_{ij}} = \mathrm{MLP}_{\boldsymbol{\phi}}(\boldsymbol{e}_{ij}). \tag{23}
$$

Next, we perform $K$ graph convolution layers, with each layer in the form of two interleaved half-convolutions. The convolution layer is defined as follows:

$$
\begin{aligned}
\boldsymbol{h}_{v_i}^{(k+1)} &\leftarrow \mathrm{MLP}_{\boldsymbol{\phi}} \left( \boldsymbol{h}_{v_i}^{(k)}, \sum_{j:e_{ij} \in \mathcal{E}} \mathrm{MLP}_{\boldsymbol{\phi}} \left( \boldsymbol{h}_{v_i}^{(k)}, \boldsymbol{h}_{e_{ij}}, \boldsymbol{h}_{v_j}^{(k)} \right) \right), \\
\boldsymbol{h}_{w_j}^{(k+1)} &\leftarrow \mathrm{MLP}_{\boldsymbol{\phi}} \left( \boldsymbol{h}_{w_j}^{(k)}, \sum_{i:e_{ij} \in \mathcal{E}} \mathrm{MLP}_{\boldsymbol{\phi}} \left( \boldsymbol{h}_{v_i}^{(k+1)}, \boldsymbol{h}_{e_{ij}}, \boldsymbol{h}_{w_j}^{(k)} \right) \right).
\end{aligned} \tag{24}
$$

The convolution layer is followed by two GraphNorm layers, one for constraint vertices and the other for variable vertices. We employ a concatenation Jumping Knowledge layer to aggregate information from all $K$ layers and obtain the node representations:

$$\boldsymbol{h}_{v_i} = \text{MLP}_{\boldsymbol{\phi}}\left(\underset{k=0,\cdots,K}{\text{CONCAT}}\left(\boldsymbol{h}_{v_i}^{(k)}\right)\right), \quad \boldsymbol{h}_{w_j} = \text{MLP}_{\boldsymbol{\phi}}\left(\underset{k=0,\cdots,K}{\text{CONCAT}}\left(\boldsymbol{h}_{w_j}^{(k)}\right)\right). \tag{25}$$

The obtained representations contain information about the instances. Subsequently, we use two MLPs to output the mean and log variance, and then sample the latent vectors for each vertex from a Gussian distribution as follows:

$$\begin{aligned}
\boldsymbol{z}_{v_i} &\sim \mathcal{N}\left(\text{MLP}_{\boldsymbol{\phi}}\left(\boldsymbol{h}_{v_i}\right), \exp \text{MLP}_{\boldsymbol{\phi}}\left(\boldsymbol{h}_{v_i}\right)\right), \\
\boldsymbol{z}_{w_j} &\sim \mathcal{N}\left(\text{MLP}_{\boldsymbol{\phi}}\left(\boldsymbol{h}_{w_j}\right), \exp \text{MLP}_{\boldsymbol{\phi}}\left(\boldsymbol{h}_{w_j}\right)\right).
\end{aligned} \tag{26}$$

### A.3.2 Decoder

The decoder implements $p_{\boldsymbol{\theta}}(\mathcal{G}|\mathbf{Z}, \tilde{\mathcal{G}})$ in Equation 22. It utilizes a GNN structure to obtain the representations, which has the same structure as the encoder GNN, but is with parameters $\boldsymbol{\theta}$ instead of $\boldsymbol{\phi}$. To encode the masked graph, we assign a special [mask] token to the masked vertex $\tilde{v}$. Its initial embedding $\boldsymbol{h}_{\tilde{v}}^{(0)}$ is initialized as a special embedding $\boldsymbol{h}_{\text{[mask]}}$. We mask all edges between $\tilde{v}$ and the variable vertices and add virtual edges. In each convolution layer, we apply a special update rule for $\tilde{v}$:

$$\boldsymbol{h}_{\tilde{v}}^{(k+1)} \leftarrow \text{MLP}_{\boldsymbol{\theta}}\left(\boldsymbol{h}_{\tilde{v}}^{(k)}, \underset{w_j \in \mathcal{W}}{\text{MEAN}}\left(\boldsymbol{h}_{w_j}^{(k+1)}\right)\right), \quad \boldsymbol{h}_{w_j}^{(k+1)} \leftarrow \text{MLP}_{\boldsymbol{\theta}}\left(\boldsymbol{h}_{w_j}^{(k+1)}, \boldsymbol{h}_{\tilde{v}}^{(k+1)}\right). \tag{27}$$

This updating is performed after each convolution layer, allowing $\tilde{v}$ to aggregate and propagate the information from the entire graph.

The obtained representations are used for the four networks—Bias Predictor, Degree Predictor, Logits Predictor, and Weights Predictor—to determine the generated graph. The details of these networks have been described in the main paper. Here we provide the losses for the four prediction tasks. In the following context, the node features, e.g., $\boldsymbol{h}_{\tilde{v}}$, refer to those from $\tilde{\mathcal{G}}$ obtained by the decoder GNN.

① Bias Prediction Loss:

$$\mathcal{L}_1(\boldsymbol{\theta}, \boldsymbol{\phi}|\mathcal{G}, \tilde{\mathcal{G}}) = \text{MSE}\left(\sigma\left(\text{MLP}_{\boldsymbol{\theta}}^{\text{bias}}\left([\boldsymbol{h}_{\tilde{v}}, \boldsymbol{z}_{\tilde{v}}]\right)\right), b_{\tilde{v}}^*\right). \tag{28}$$

② Degree Prediction Loss:

$$\mathcal{L}_2(\boldsymbol{\theta}, \boldsymbol{\phi}|\mathcal{G}, \tilde{\mathcal{G}}) = \text{MSE}\left(\sigma\left(\text{MLP}_{\boldsymbol{\theta}}^{\text{deg}}\left([\boldsymbol{h}_{\tilde{v}}, \boldsymbol{z}_{\tilde{v}}]\right)\right), d_{\tilde{v}}^*\right). \tag{29}$$

③ Logits Prediction Loss:

$$\begin{aligned}
\mathcal{L}_3(\boldsymbol{\theta}, \boldsymbol{\phi}|\mathcal{G}, \tilde{\mathcal{G}}) &= -\mathbb{E}_{(\tilde{v},u)\sim p_{\text{pos}}}\left[\log\left(\hat{\delta}_{\tilde{v},u}'\right)\right] - \mathbb{E}_{(\tilde{v},u)\sim p_{\text{neg}}}\left[\log\left(1 - \hat{\delta}_{\tilde{v},u}'\right)\right], \\
\hat{\delta}_{\tilde{v},u}' &= \sigma\left(\text{MLP}_{\boldsymbol{\theta}}^{\text{logits}}\left([\boldsymbol{h}_u, \boldsymbol{z}_u]\right)\right).
\end{aligned} \tag{30}$$

④ Weights Prediction Loss:

$$\mathcal{L}_4(\boldsymbol{\theta}, \boldsymbol{\phi}|\mathcal{G}, \tilde{\mathcal{G}}) = \text{MSE}\left(\sigma\left(\text{MLP}_{\boldsymbol{\theta}}^{\text{weights}}\left([\boldsymbol{h}_u, \boldsymbol{z}_u]\right)\right), e_{\tilde{v},u}^*\right). \tag{31}$$

With these four prediction tasks, the reconstruction loss in Equation 22 is instantiated as:

$$\mathcal{L}_{\text{rec}}(\boldsymbol{\theta}, \boldsymbol{\phi}|\mathcal{G}, \tilde{\mathcal{G}}) = \sum_{i=1}^{4} \alpha_i \cdot \mathcal{L}_i(\boldsymbol{\theta}, \boldsymbol{\phi}|\mathcal{G}, \tilde{\mathcal{G}}), \tag{32}$$

### A.3.3 Training and Inference

We describe the training and inference procedures in Algorithm 1 and Algorithm 2, respectively.

---
**Algorithm 1:** Train G2MILP
---
**Input:** Dataset $\mathcal{D}$, number of training steps $N$, batch size $B$.
**Output:** Trained G2MILP, dataset statistics $\underline{b}, \overline{b}, \underline{d}, \overline{d}, \underline{e}, \overline{e}$.

1  Calculate the statistics $\underline{b}, \overline{b}, \underline{d}, \overline{d}, \underline{e}, \overline{e}$ over $\mathcal{D}$;
2  **for** $n = 1, \cdots, N$ **do**
3      $\mathcal{B} \leftarrow \emptyset$;
4      **for** $b = 1, \cdots, B$ **do**
5          $\mathcal{G} \sim \mathcal{D}, \tilde{v} \sim \mathcal{V}^{\mathcal{G}}$;
6          $\tilde{\mathcal{G}} \leftarrow \mathcal{G}.\text{MaskNode}(\tilde{v})$;
7          $\mathcal{B} \leftarrow \mathcal{B} \cup \{(\mathcal{G}, \tilde{\mathcal{G}})\}$;
8          Compute $b_{\tilde{v}}^*, d_{\tilde{v}}^*, \delta_{\tilde{v},u}, e_{\tilde{v},u}^*$;
9          Compute $\mathcal{L}(\boldsymbol{\theta}, \boldsymbol{\phi}|\mathcal{G}, \tilde{\mathcal{G}})$ in Equation 22;
10      $\mathcal{L}(\boldsymbol{\theta}, \boldsymbol{\phi}) \leftarrow \frac{1}{|\mathcal{B}|} \sum_{(\mathcal{G}, \tilde{\mathcal{G}}) \in \mathcal{B}} \mathcal{L}(\boldsymbol{\theta}, \boldsymbol{\phi}|\mathcal{G}, \tilde{\mathcal{G}})$;
11      Update $\boldsymbol{\theta}, \boldsymbol{\phi}$ to minimize $\mathcal{L}(\boldsymbol{\theta}, \boldsymbol{\phi})$.

---
**Algorithm 2:** Generate a MILP instance
---
**Input:** Dataset $\mathcal{D}$, trained G2MILP, dataset statistics $\underline{b}, \overline{b}, \underline{d}, \overline{d}, \underline{e}, \overline{e}$, masking ratio $\eta$.
**Output:** A novel instance $\hat{\mathcal{G}}$.

1  $\mathcal{G} \sim \mathcal{D}, N_{\text{iters}} \leftarrow \eta \cdot |\mathcal{V}^{\mathcal{G}}|, \hat{\mathcal{G}} \leftarrow \mathcal{G}$;
2  **for** $n = 1, \cdots, N_{iters}$ **do**
3      $\tilde{v} \sim \mathcal{V}^{\hat{\mathcal{G}}}$;
4      Compute $\hat{b}_{\tilde{v}}^*, \ \hat{b}_{\tilde{v}} \leftarrow \underline{b} + (\overline{b} - \underline{b}) \cdot \hat{b}_{\tilde{v}}^*, \ \tilde{\mathcal{G}}.\tilde{v}.\text{bias} \leftarrow \hat{b}_{\tilde{v}}$;
5      Compute $\hat{d}_{\tilde{v}}^*, \ \hat{d}_{\tilde{v}} \leftarrow \underline{d} + (\overline{d} - \underline{d}) \cdot \hat{d}_{\tilde{v}}^*$;
6      **for** $u \in \mathcal{W}^{\tilde{\mathcal{G}}}$ **do**
7          Compute $\hat{\delta}'_{\tilde{v},u}$;
8      **for** $u \in \arg \text{TopK}(\{\hat{\delta}'_{\tilde{v},u}|u \in \mathcal{W}^{\tilde{\mathcal{G}}}\}, \hat{d}_{\tilde{v}})$ **do**
9          Compute $\hat{e}_{\tilde{v},u}^*, \ \hat{e}_{\tilde{v},u} \leftarrow \underline{e} + (\overline{e} - \underline{e}) \cdot \hat{e}_{\tilde{v},u}^*$;
10          $\tilde{\mathcal{G}}.\text{AddEdge}(\tilde{v}, u)$;
11          $\tilde{\mathcal{G}}.e_{\tilde{v},u}.\text{weights} \leftarrow \hat{e}_{\tilde{v},u}$;
12      $\hat{\mathcal{G}} \leftarrow \tilde{\mathcal{G}}$;
13  Output $\hat{\mathcal{G}}$.

---

# B  Experimental Details

## B.1  Dataset

The three commonly used datasets, namely MIS, SetCover, and MIK, are the same as those used in [9]. Nurse Scheduling contains a group of $4$ instances from MI-PLIB 2017: `nursesched-medium04` and `nursesched-sprint-hidden09` for training, and `nursesched-sprint02` and `nursesched-sprint-late03` for test. Table 6 summarizes some statistics of these datasets.

Table 6: Statistics of datasets. Size means the number of instances in the training set. $|\mathcal{V}|$ and $|\mathcal{W}|$ are the numbers of constraints and variables, respectively.

| Dataset | MIS | SetCover | MIK | Nurse Scheduling |
|---|---|---|---|---|
| Size | 1000 | 1000 | 80 | 2 |
| Mean $|\mathcal{V}|$ | 1953 | 500 | 346 | 8707 |
| Mean $|\mathcal{W}|$ | 500 | 1000 | 413 | 20659 |

## B.2 Hyperparameters

We report some important hyperparameters in this section. Further details can be found in our code once the paper is accepted to be published.

We run our model on a single GeForce RTX 3090 GPU. The hidden dimension and the embedding dimension are set to 16. The depth of the GNNs is 6. Each MLP has one hidden layer and uses `ReLU()` as the activation function.

In this work, we simply set all $\alpha_i$ to 1. We find that the choice of $\beta$ significantly impacts the model performance. For MIS, we set $\beta$ to 0.00045. For SetCover, MIK and Nurse Scheduling, we apply a sigmoid schedule [46] to let $\beta$ to reach 0.0005, 0.001, and 0.001, respectively. We employ the Adam optimizer, train the model for $20,000$ steps, and choose the best checkpoint based on the average error in solving time and the number of branching nodes. The learning rate is initialized to 0.001 and decays exponentially. For MIS, SetCover, and MIK, we set the batch size to 30. Specifically, to provide more challenging prediction tasks in each batch, we sample 15 graphs and use each graph to derive 2 masked ones for training. For Nurse Scheduling, we set the batch size as 1 due to the large size of each graph.

## B.3 Structural Distributional Similarity

Table 7: Description of statistics used for measuring the structural distributional similarity. These statistics are calculated on the bipartite graph extracted by Ecole.

| Feature | Description |
| --- | --- |
| coef_dens | Fraction of non-zero entries in $\boldsymbol{A}$, i.e., $\|\mathcal{E}\|/(\|\mathcal{V}\| \cdot \|\mathcal{W}\|)$. |
| cons_degree_mean | Mean degree of constraint vertices in $\mathcal{V}$. |
| cons_degree_std | Std of degrees of constraint vertices in $\mathcal{V}$. |
| var_degree_mean | Mean degree of variable vertices in $\mathcal{W}$. |
| var_degree_std | Std of degrees of variance vertices in $\mathcal{W}$. |
| lhs_mean | Mean of non-zero entries in $\boldsymbol{A}$. |
| lhs_std | Std of non-zero entries in $\boldsymbol{A}$. |
| rhs_mean | Mean of $\boldsymbol{b}$. |
| rhs_std | Std of $\boldsymbol{b}$. |
| clustering_coef | Clustering coefficient of the graph. |
| modularity | Modularity of the graph. |

Table 7 presents the 11 statistics that we use to measure the structural distributional similarity. First, we calculate the statistics for each instance. We then compute the JS divergence $D_{\text{JS},i}$ between the generated samples and the training set for each descriptor $i \in \{1, \cdots, 11\}$. We estimate the distributions using the histogram function in numpy and the cross entropy using the entropy function in scipy. The JS divergence falls in the range $[0, \log 2]$, so we standardize it to a score $s_i$ via:

$$s_i = \frac{1}{\log 2} \left( \log 2 - D_{\text{JS},i} \right). \tag{33}$$

Then we compute the mean of the 11 scores for the descriptors to obtain the final score $s$:

$$s = \frac{1}{11} \sum_{i=1}^{11} s_i. \tag{34}$$

Hence the final score ranges from 0 to 1, with a higher score indicating better similarity.

We use the training set to train a G2MILP model for each dataset and generate 1000 instances to compute the similarity scores. For MIK, which has only 80 training instances, we estimated the score using put-back sampling.

Table 8: Results on the optimal value prediction task (mean±std). On each dataset and for each method, we sample 5 different sets of 20 instances for augmentation.

| | MIK | | Nurse Scheduling | |
|---|---|---|---|---|
| | MSE | Improvement | MSE | Improvement |
| Dataset | 0.0236 | 0.0% | 679.75 | 0.0% |
| Bowly | - | - | 663.52 (±95.33) | 2.3% (±14.0%) |
| Random | 0.0104 (±0.0023) | 55.9% (±9.7%) | - | - |
| G2MILP | 0.0073 (±0.0014) | 69.1% (±5.9%) | 548.70 (±44.68) | 19.3% (±6.6%) |

Table 9: Results on the predict-and-search framework on MIS. The training set contains 100 instances, and we generate 100 new instances. For Random and G2MILP, masking ratio is 0.01. Time means the time for Gurobi to find the optimal solution with augmenting data generated by different models. Bowly leads to the framework failing to find optimal solutions in the trust region.

| Method | Training Set | Bowly | Random | G2MILP |
|---|---|---|---|---|
| **Time** | 0.041 (±0.006) | 17/100 fail | 0.037 (±0.003) | 0.032 (±0.004) |

Notice that for a fair comparison, we exclude statistics that remain constant in our approach, such as problem size and objective coefficients. We implement another version of metric that involves more statistics, and the results are in Appendix C.3.

## B.4 Downstream Tasks

The generated instances have the potential to enrich dataset in any downstream task. In this work, we demonstrate this potential through two application scenarios, i.e., the optimal value prediction task and the predict-and-search framework.

**Optimal Value Prediction** Two datasets, MIK and Nurse Scheduling, are considered, with medium and extremely small sizes, respectively. Following [18], we employ a GNN as a predictive model. The GNN structure is similar to the GNNs in G2MIL. We obtain the graph representation using mean pooling over all vertices, followed by a two-layer MLP to predict the optimal values of the instances.

For each dataset, we train a GNN predictive model on the training set. Specifically, for MIK, we use 80% of instances for training, 20% of instances for validating, and train for 1000 epochs to select the best checkpoint based on validation MSE. For Nurse Scheduling, we use both instances to train the model for 80 epochs. We use the generative models, Bowly, Random, and G2MILP, to generate 20 instances similar to the training sets. For Random and G2MILP, we mix together the instances generated by setting the masking ratio $\eta$ to 0.01 and 0.05, respectively. Next, we use the generated instances to enrich the original training sets, and use the enriched data to train another predictive model. We test all the trained model on previously unseen test data. Table 8 presents the predictive MSE on the test sets of the models trained on different training sets. As the absolute values of MSE are less meaningful than the relative values, we report the performance improvements brought by different generative technique. The improvement of $Model_2$ relative to $Model_1$ is calculate as follows:

$$\text{Improvement}_{2,1} = \frac{\text{MSE}_1 - \text{MSE}_2}{\text{MSE}_1}. \tag{35}$$

On MIK, Bowly results in numerical issues as some generated coefficients are excessively large. G2MILP significant improves the performance and outperforms Random. On Nurse Scheduling, Random fails to generate feasible instances, and Bowly yields a minor improvement. Notably, G2MILP allows for the training of the model with even minimal data.

**Predict-and-Search** We conduct experiments on a neural solver, i.e., the predict-and-search framework proposed by Han et al. [31] Specifically, they propose a framework that first predicts a solution and then uses solvers to search for the optimal solutions in a trust region. We consider using generated instances to enhance the predictive model. We first train the predictive model on 100 MIS instances, and then use the generative models to generate 100 new instances to augment the dataset. The results

are in Table 9 . Bowly generates low-quality data that disturbs the model training, so that there is no optimal solution in the trust region around the predicted solution. Though both Random and G2MILP can enhance the solving framework to reduce solving time, we can see G2MILP significantly outperforms Random.

**Discussions**    These two downstream tasks, despite their simplicity, possess characteristics that make them representative problems that could benefit from generative models. Specifically, we identify the following features.

1. **More is better.** We want as many data instances as possible. This condition is satisfied when we can obtain precise labels using existing methods, e.g., prediction-based neural solvers [31], or when unlabeled data is required for RL model training, e.g., RL for cut selection [9].

2. **More similar is better.** We want independent identically distributed (i.i.d.) data instances for training. Thus

3. **More diverse is better.** We want the data to be diverse, despite being i.i.d., so that the trained model can generalize better.

Our experimental results demonstrate the potential of G2MILP in facilitating downstream tasks with these characteristics, thus enhancing the MILP solvers. We intend to explore additional application scenarios in future research.

# C    Additional Results

## C.1    Comparison with G2SAT

Table 10:  Results of G2SAT on MIS. In the table, "sim" denotes similarity score (higher is better), "time" denotes solving time, and "#branch" denotes the number of branching nodes, respectively. Numbers in brackets denote relative errors (lower is better).

|  | sim | time (s) | #branch |
|---|---|---|---|
| Training Set | 0.998 | 0.349 | 16.09 |
| G2SAT | 0.572 | 0.014 (96.0%) | 2.11 (86.9%) |
| G2MILP ($\eta = 0.01$) | 0.997 | 0.354 (1.5%) | 15.03 (6.6%) |
| G2MILP ($\eta = 0.1$) | 0.895 | 0.214 (38.7%) | 4.61 (71.3%) |

We conduct an additional experiment that transfers G2SAT to a special MILP dataset, MIS, in which all coefficients are $1.0$ and thus the instances can be modeled as homogeneous bipartite graphs. We apply G2SAT to learn to generate new graphs and convert them to MILPs. Results are in Table 10. The results show that G2MILP significantly outperforms G2SAT on the special cases.

## C.2    Masking Process

**Masking Variables**    In the mainbody, for simplicity, we define the masking process of uniformly sampling a constraint vertex $\tilde{v} \sim \mathcal{U}(\mathcal{V})$ to mask, while keeping the variable vertices unchanged. We implement different versions of G2MILP that allow masking and modifying either constraints, variables, or both. The results are in Table 11.

**Ablation on Masking Ratio**    We have conduct ablation studies on the effect of the masking ratio $\eta$ on MIK. The results are in Figure 4. The experimental settings are the same with those in Table 2 and Figure 2. From the results we have the following conclusions. (1) though empirically a smaller leads to a relatively better performance, G2MILP maintains a high similarity performance even when $\eta$ is large. (2) The downstream task performance does not drops significantly. This makes sense because smaller $\eta$ leads to more similar instances, while larger $\eta$ leads to more diverse (but still similar) instances, both of which can benefit downstream tasks. (3) G2MILP always outperforms Random,

Table 11: Results of different implementations of G2MILP on MIK. In the table, $\eta$ denotes mask ratio, "v" denotes only modifying variables (objective coefficients and variable types), "c" denotes only modifying constraints, and "vc" denotes first modifying variables and then modifying constraints. We do not report the similarity scores for "v" models because the current similarity metrics exclude statistics that measure only variables.

|  |  | sim | time (s) | #branch |
|---|---|---|---|---|
| Training Set |  | 0.997 | 0.198 | 175.35 |
| G2MILP ($\eta = 0.01$) | v | - | 0.183 (7.5%) | 136.68 (22.0%) |
|  | c | 0.989 | 0.169 (17.1%) | 167.44 (4.5%) |
|  | vc | 0.986 | 0.186 (6.1%) | 155.40 (11.4%) |
| G2MILP ($\eta = 0.05$) | v | - | 0.176 (11.1%) | 136.68 (22.0%) |
|  | c | 0.964 | 0.148 (25.3%) | 150.90 (13.9%) |
|  | vc | 0.964 | 0.147 (25.3%) | 142.70 (18.6%) |
| G2MILP ($\eta = 0.1$) | v | - | 0.172 (13.1%) | 136.67 (22.1%) |
|  | c | 0.905 | 0.117 (40.9%) | 169.63 (3.3%) |
|  | vc | 0.908 | 0.115 (41.9%) | 112.29 (35.9%) |

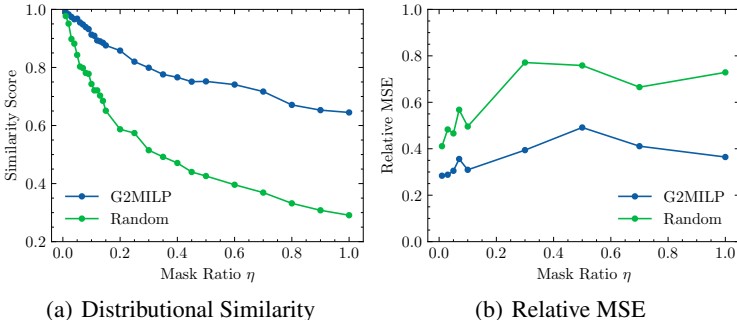

(a) Distributional Similarity      (b) Relative MSE

Figure 4: (a) Distributional similarity score (higher is better) and (b) Relative MSE (lower is better) v.s. masking ratio $\eta$.

which demonstrates that the learning paradigm helps maintain the performance. (4) Bowly fails on this dataset because its generated instances lead to numerical issues and cannot be read by Gurobi or SCIP. Moreover, in real applications, it is reasonable and flexible to adjust the hyperparameter to achieve good performances in different scenarios.

**Orders of Masked Constraints** We also investigate different orders of masking constraint vertices, including uniformly sampling and sampling according to the vertex indices. Results are in Table 12. We find that uniformly sampling achieves the best performance. Sampling according to indices leads to a performance decrease, maybe because near constraints are relevant and lead to error accumulation. We think these results are interesting, and will study it in the future work.

### C.3 Structural Distributional Similarity

In the mainbody, for a fair comparison, we exclude statistics that remain constant in our approach, such as problem size and objective coefficients. However, these statistics are also important features for MILPs. In this section, we incorporate three additional statistics in the computing of similarity scores: (1) mean of objective coefficients $\mathbf{c}$, (2) std of objective coefficients $\mathbf{c}$, and (3) the ratio of continuous variables. With these additional metrics, we recompute the structural similarity scores and updated the results in both Table 2 and Table 11. The new results are in Table 13 and Table 14,

Table 12: Results of different implementations of generation orders on MIK dataset. In the table, "Uni" denotes uniformly sampling from constraints. "Idx ↗" and "Idx ↘" denote sampling constraints according to indices in ascending order and descending order, respectively.

| order | model | sim | time (s) | #branch |
|-------|-------|-----|----------|---------|
| Uni | G2MILP | 0.953 | 0.129 (35.1%) | 235.35 (34.2%) |
| | Random | 0.840 | 0.004 (97.9%) | 0.00 (100%) |
| Idx ↗ | G2MILP | 0.892 | 0.054 (72.7%) | 108.30 (38.2%) |
| | Random | 0.773 | 0.002 (98.9%) | 0.00 (100%) |
| Idx ↘ | G2MILP | 0.925 | 0.027 (86.2%) | 31.53 (82.0%) |
| | Random | 0.827 | 0.003 (98.6%) | 0.00 (100%) |

respectively. From the results, we can still conclude that G2MILP outperforms all baselines, further supporting the effectiveness of our proposed method.

Table 13: (Table 2 recomputed.) Structural distributional similarity scores between the generated instances with the training datasets. Higher is better. $\eta$ is the masking ratio. We do not report the results of Bowly on MIK because Ecole [45] and SCIP [51] fail to read the generated instances due to large numerical values.

| | | MIS | SetCover | MIK |
|---|---|-----|----------|-----|
| | Bowly | 0.144 | 0.150 | - |
| $\eta = 0.01$ | Random | 0.722 | 0.791 | 0.971 |
| | G2MILP | **0.997** | **0.874** | **0.994** |
| $\eta = 0.05$ | Random | 0.670 | 0.704 | 0.878 |
| | G2MILP | **0.951** | **0.833** | **0.969** |
| $\eta = 0.1$ | Random | 0.618 | 0.648 | 0.768 |
| | G2MILP | **0.921** | **0.834** | **0.930** |

Table 14: (Table 11 recomputed.) Results of different implementations of G2MILP on MIK. In the table, $\eta$ denotes mask ratio, "v" denotes only modifying variables (objective coefficients and variable types), "c" denotes only modifying constraints, and "vc" denotes first modifying variables and then modifying constraints.

| | v | c | cv |
|---|---|---|----|
| G2MILP ($\eta = 0.01$) | 0.998 | 0.988 | 0.985 |
| G2MILP ($\eta = 0.05$) | 0.996 | 0.968 | 0.967 |
| G2MILP ($\eta = 0.1$) | 0.996 | 0.928 | 0.912 |

## C.4 Sizes of Datasets

We conduct experiments on different sizes of the original datasets, as well as the ratio of generated instances to original ones, on MIS. The results are in Table 15. The results show that G2MILP can bring performance improvements on varying sizes of datasets.

## C.5 Visualization

The t-SNE visualization for baselines are in Figure 5. G2MILP generates diverse instances around the training set, while instances generated by Random are more biased from the realistic ones.

Table 15: Results on the optimal value prediction task on MIS with different dataset sizes. In the table, "#MILPs" denotes the number of instances in the training sets, and "Augment%" denotes the ratio of generated instances to training instances.

| #MILPs | Augment% | MSE | Improvement |
|---|---|---|---|
| | 0 | 1.318 | 0 |
| 50 | 25% | 1.014 | 23.1% |
| | 50% | 0.998 | 24.3% |
| | 100% | 0.982 | 25.5% |
| | 0 | 0.798 | 0 |
| 100 | 25% | 0.786 | 1.5% |
| | 50% | 0.752 | 5.8% |
| | 100% | 0.561 | 23.7% |
| | 0 | 0.294 | 0 |
| 200 | 25% | 0.283 | 19.0% |
| | 50% | 0.243 | 17.3% |
| | 100% | 0.202 | 31.3% |
| | 0 | 0.188 | 0 |
| 500 | 25% | 0.168 | 10.6% |
| | 50% | 0.175 | 6.9% |
| | 100% | 0.170 | 9.6% |

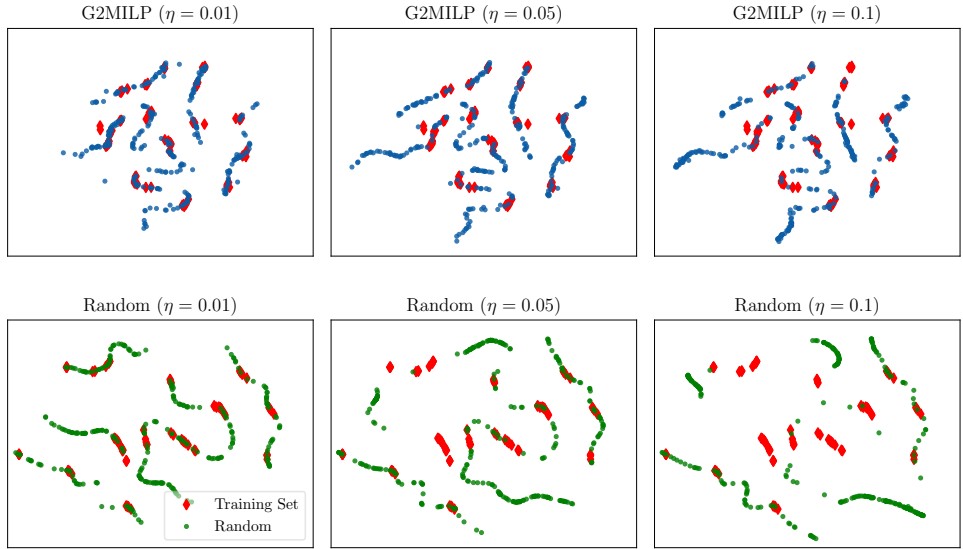

Figure 5: The t-SNE visualization of MILP instance representations for MIK. Each point represents an instance. Red points are from the training set, blue points are instances generated by G2MILP, and green points are instances generated by Random.

