| Bowly | | $0.007 \pm 0.00$ (97.9%) | $0.048 \pm 0.00$ (97.9%) | $0.001 \pm 0.00$ (99.8%) |
| $\eta = 0.01$ | Random | $0.311 \pm 0.05$ (10.8%) | $2.044 \pm 0.19$ (12.8%) | $0.008 \pm 0.00$ (96.1%) |
| | G2MILP | **$0.354 \pm 0.06$ (1.5%)** | **$2.360 \pm 0.18$ (0.8%)** | **$0.169 \pm 0.07$ (14.7%)** |
| $\eta = 0.05$ | Random | $0.569 \pm 0.09$ (63.0%) | $2.010 \pm 0.11$ (14.3%) | $0.004 \pm 0.00$ (97.9%) |
| | G2MILP | **$0.292 \pm 0.07$ (16.3%)** | **$2.533 \pm 0.15$ (8.1%)** | **$0.129 \pm 0.05$ (35.1%)** |
| $\eta = 0.1$ | Random | $2.367 \pm 0.35$ (578.2%) | $1.988 \pm 0.17$ (15.2%) | $0.005 \pm 0.00$ (97.6%) |
| | G2MILP | **$0.214 \pm 0.05$ (38.7%)** | **$2.108 \pm 0.21$ (10.0%)** | **$0.072 \pm 0.02$ (63.9%)** |