# OpenReview forum: "A Deep Instance Generative Framework for MILP Solvers Under Limited Data Availability"
_NeurIPS.cc/2023/Conference — NeurIPS 2023 spotlight_

### Official Review · Reviewer_sm9N · 2023-07-01

**Soundness:** 3 good
**Presentation:** 3 good
**Contribution:** 3 good
**Rating:** 7
**Confidence:** 3

**Summary:**

The paper discusses the recent surge in using machine learning techniques for solving combinatorial optimization problems, particularly mixed-integer linear programs (MILPs). However, the limited availability of real-world instances often hinders optimal decision-making and unbiased solver assessments. To address this issue, the proposed solution is G2MILP, a deep generative framework for MILP instances. G2MILP represents MILP instances as bipartite graphs and employs a masked variational autoencoder to corrupt and replace parts of the graphs iteratively, generating new instances. This approach learns to generate realistic MILP instances without relying on expert-designed formulations, while preserving the structures and computational complexity of real-world datasets. The generated instances can assist in improving MILP solvers when data availability is limited. A set of benchmarks is designed to evaluate the quality of the generated MILP instances, and experimental results demonstrate their resemblance to real-world datasets in terms of both structure and computational difficulty.

**Strengths:**

1. The paper takes a traditionally hard problem and proposes a novel solution to it.
2. Experiment results seem very compellingly, however the lack of good baseline makes hard to judge how good it actually is.
3. Presentation is good and clear.

**Weaknesses:**

some minor issue might be lack of proper baseline, it's quite hard for the readers and reviewers to access how good this method actually is.

**Questions:**

Is it possible to provide some "fair" comparisons on downstream proxy tasks with other popular methods? I must confess that I am not very familiar with the specific subdomain.

---

> ### Author Rebuttal · Authors · 2023-08-09
>
> We thank the reviewer for the insightful and valuable comments. We respond to your comments as follows and sincerely hope that our rebuttal could properly address your concerns. If so, we would deeply appreciate it if you could raise your score and your confidence. If not, please let us know your further concerns, and we will continue actively responding to your comments and improving our work.
>
> **Weakness.**
>
> > some minor issue might be lack of proper baseline, it's quite hard for the readers and reviewers to access how good this method actually is.
>
> - Thanks for your comment! **We want to emphasize that G2MILP, to the best of our knowledge, is the first learning based method to generate realistic MILP instances**. We open up a new research direction while previous methods cannot deal with such a task. Therefore, we can hardly find strong existing baselines.
>
> - **Nonetheless, we have delicately designed two baselines, i.e., Bowly and Random, for a fair comparison.**
>
>   - **Bowly** is a traditional rule-based MILP instance generation technique. It generates instances from scratch by randomly sampling the coefficients. It was not designed for learning datasets to generate realistic instances, and we can only tune simple statistics such as problem sizes and coefficient means. For a fair comparison, we keep all controllable parameters  the same as the training sets. Unsurprisingly, it does not work well on the new task.
>   - We design the **Random** baseline as a comparison to demonstrate the advantage of G2MILP, especially the deep learning modules. The comparison with this baseline demonstrates that modifying MILPs while reserving both graph semantic structure and difficulty is nontrivial, and our proposed deep learning framework is able to achieve that.
>
> - **For a better comparison, we add an additional experiment to compare G2MILP with G2SAT.**  Specifically, we adapt G2SAT to a special MILP dataset, MIS, in which all coefficients are 1.0 and thus the instances can be modeled as homogeneous bipartite graphs. Notice that G2SAT can only be applied on homogenous bipartite graphs, and so it only works on such special MILP data. We apply G2SAT to learn to generate new graphs and convert them to MILPs. Results are in **Table 9** in the newly submitted PDF. The results show that G2MILP significantly outperforms G2SAT on the special cases. For your convenience, we quote Table 9 here.
>
>
>   **Table 9: Results of G2SAT on MIS. In the table, ''sim'' denotes similarity score (higher is better), ''time'' denotes solving time, and ''\#branch'' denotes the number of branching nodes, respectively. Numbers in brackets denote relative errors (lower is better).**
>
>   |                      | sim   | time (s)      | #branch      |
>   | -------------------- | ----- | ------------- | ------------ |
>   | Training Set         | 0.998 | 0.349         | 16.09        |
>   | G2SAT                | 0.572 | 0.014 (96.0%) | 2.11 (86.9%) |
>   | G2MILP ($\eta=0.01$) | 0.997 | 0.354 (1.5%)  | 15.03 (6.6%) |
>   | G2MILP ($\eta=0.1$)  | 0.895 | 0.214 (38.7%) | 4.61 (71.3%) |
>
> **Question.**
>
> > Is it possible to provide some "fair" comparisons on downstream proxy tasks with other popular methods?
>
> - Thanks for your suggestion! Previous popular methods include **instance generation methods** and **MILP solving methods**. For the former, though they are not designed for the  learning to generate task, we have designed the baselines and conduct comprehensive comparisons. For the latter, notice that G2MILP is orthogonal to them and can be used to enhance them. We also consider the downstream task (optimal value prediction) to demonstrate the effectiveness of G2MILP.
>
> - **For further demonstration, we conduct an additional downstream task, i.e., the predict-and-search framework proposed in [1].** This paper proposes a framework that first predicts a solution and then uses solvers to search for the optimal solutions in a trust region. We consider using generated instances to enhance the predictive model. We first train the predictive model on 100 MIS instances, and then use the generative models to generate 100 new instances to augment the dataset. The results are in **Table 14**. The results show that G2MILP can effectively enhance previous popular methods. For your convenience, we quote Table14 here.
>
>   **Table 14: Results on the predict-and-search framework on MIS. The training set contains 100 instances, and we generate 100 new instances. For Random and G2MILP, masking ratio is 0.01. Time means the time for Gurobi to find the optimal solution with augmenting data generated by different models. Bowly leads to the framework failing to find optimal solutions in the trust region.**
>
>   | Method |    Training Set     |    Bowly    |       Random        |       G2MILP        |
>   | :----: | :-----------------: | :---------: | :-----------------: | :-----------------: |
>   |  Time  | 0.041 ($\pm$ 0.006) | 17/100 fail | 0.037 ($\pm$ 0.003) | 0.032 ($\pm$ 0.004) |
>
> [1] Han Q, Yang L, Chen Q, et al. A GNN-Guided Predict-and-Search Framework for Mixed-Integer Linear Programming. ICLR, 2023.

---

> > ### Comment · Reviewer_sm9N · 2023-08-17
> > **Increasing score to accept**
> >
> > Thank you for your effort meticulously addressing other reviewers' comments and mine. After reading all the discussions above, I believe this work is significant and should be presented to the community. Hence raising the score to accept.

---

> > > ### Author Response · Authors · 2023-08-17
> > > **Thank you for your kind support.**
> > >
> > > Dear Reviewer sm9N,
> > >
> > > Thanks for your kind support and for helping us improve the paper! We appreciate your valuable suggestions.
> > >
> > > Best,
> > >
> > > Authors

---

### Official Review · Reviewer_Vxmz · 2023-07-05

**Soundness:** 3 good
**Presentation:** 4 excellent
**Contribution:** 3 good
**Rating:** 7
**Confidence:** 4

**Summary:**

The authors propose a deep generative framework for MILP(mixed integer programs) instances. The work has applications in enhancing MILP solvers under limited data availability.  MILP instances are represented as weighted bipartite graphs.  The authors propose a masked  variational autoencoder (VAE) method for graph generation.

The problem is relevant and interesting. The experiments show that the proposed approach obtains significantly better results than existing methods.

**Strengths:**

1.  Important problem to tackle. The work has applications.

2. Paper is easy to understand.

3. The authors motivate different components of encoder decoder architecture nicely in the methodology section.

4. The experimental section consists of Structural similarity metrics, hardness comparison of generated MILP instances, application on downstream task.

5. Comparison on variety of datasets( MIS, Set cover, MIK(knapsack), including standard deviation, multiple runs.

6. Visualization of generated instances( through TSNE plots).



**Weaknesses:**

1.  Downstream Task( learning to solve MIP)- learning to branch etc. :

Recently many neural MIP solvers have come up( Eg: -Gasse et al.[A]). The authors have also cited them. Is it possible for the authors to show whether they can improve the performance of such solvers by data augmentation( on any dataset) under low data availability.

I agree the authors have shown improvement on predicting optimal value task. However, the above mentioned task is also important. It can make the paper stronger.


2. Presentation :
Table 8 and Fig 2 not clear.

 Request the authors to add Standard deviation in Table 8 Appendix ( especially for Random). Also I see "Random" in Table 8 and not in Fig.2 ( Downstream task)..

Also is there any difference between Table 8 and Fig 2?


3. Code is not provided by authors.



[A] Maxime Gasse, Didier Chételat, Nicola Ferroni, Laurent Charlin, and Andrea Lodi. Exact354
combinatorial optimization with graph convolutional neural networks. Advances in neural355
information processing systems, 32, 2019

**Questions:**

Request the authors to answer the questions in weakness section.
1. Downstream task: Augmenting neural mip solvers with generated data. ( Especially under low data availability).

2. Presentation clarification( Fig 2 and Table 8).

**Limitations:**

The authors have addressed it nicely.

---

> ### Author Rebuttal · Authors · 2023-08-09
>
> We thank the reviewer for the insightful and valuable comments. We respond to your comments as follows and sincerely hope that our rebuttal could properly address your concerns. If so, we would deeply appreciate it if you could raise your score and your confidence. If not, please let us know your further concerns, and we will continue actively responding to your comments and improving our work.
>
> **Weakness 1 & Question 1.**
>
> > Recently many neural MIP solvers have come up( Eg: -Gasse et al.[A]). The authors have also cited them. Is it possible for the authors to show whether they can improve the performance of such solvers by data augmentation( on any dataset) under low data availability.
> >
> > I agree the authors have shown improvement on predicting optimal value task. However, the above mentioned task is also important. It can make the paper stronger.
>
> - Thanks for your suggestion! **We conduct an additional downstream task for a neural solver, i.e., the predict-and-search framework proposed in [B].** This paper proposes a framework that first predicts a solution and then uses solvers to search for the optimal solutions in a trust region. We consider using generated instances to enhance the predictive model. We first train the predictive model on 100 MIS instances, and then use the generative models to generate 100 new instances to augment the dataset. The results are in **Table 14**. For your convenience, we quote Table14 here.
>
>   **Table 14: Results on the predict-and-search framework on MIS. The training set contains 100 instances, and we generate 100 new instances. For Random and G2MILP, masking ratio is 0.01. Time means the time for Gurobi to find the optimal solution with augmenting data generated by different models. Bowly leads to the framework failing to find optimal solutions in the trust region.**
>
>   | Method |    Training Set     |    Bowly    |       Random        |       G2MILP        |
>   | :----: | :-----------------: | :---------: | :-----------------: | :-----------------: |
>   |  Time  | 0.041 ($\pm$ 0.006) | 17/100 fail | 0.037 ($\pm$ 0.003) | 0.032 ($\pm$ 0.004) |
>
> - The results show that Bowly generates low-quality data that disturbs the model training, so that there is no optimal solution in the trust region around the predicted solution. Though both Random and G2MILP can enhance the solving framework to reduce solving time, we can see G2MILP significantly outperforms Random, demonstrating its effectiveness.
> - We consider the predict-and-search task because it is a typical predictive task that requires large amounts of instances for training. Gasse et al.[A], however, leverages each branching decision, instead of each instance, as one data sample, so it may not be a typical application scenario of instance generative techniques.
>
> [A] Maxime Gasse, Didier Chételat, Nicola Ferroni, Laurent Charlin, and Andrea Lodi. Exact combinatorial optimization with graph convolutional neural networks. Advances in neural information processing systems, 32, 2019
>
> [B] Qingyu Han, Linxin Yang, Qian Chen, Xiang Zhou, Dong Zhang, Akang Wang, Ruoyu Sun, 410 and Xiaodong Luo. A gnn-guided predict-and-search framework for mixed-integer linear 411 programming. ICLR, 2302.
>
> **Weakness 2 & Question 2.**
>
> > Presentation clarification( Fig 2 and Table 8).
> >
> > Presentation : Table 8 and Fig 2 not clear.
> >
> > Request the authors to add Standard deviation in Table 8 Appendix ( especially for Random). Also I see "Random" in Table 8 and not in Fig.2 ( Downstream task)..
> >
> > Also is there any difference between Table 8 and Fig 2?
>
> - Thanks for your suggestion! The results with std in Table 8 are in **Table 12** in the newly submitted PDF. For your convenience, we quote Table 12 here (better viewed in the  submitted PDF).
>
>   **Table 12: Results on the optimal value prediction task (mean±std). On each dataset and for each method, we sample 5 different sets of 20 instances for augmentation.**
>
>   |         |          MIK          |          MIK           |   Nurse Scheduling   |  Nurse Scheduling   |
>   | :-----: | :-------------------: | :--------------------: | :------------------: | :-----------------: |
>   |         |          MSE          |      Improvement       |         MSE          |     Improvement     |
>   | Dataset |        0.0236         |          0.0%          |        679.75        |        0.0%         |
>   |  Bowly  |           -           |           -            | 663.52($\pm 95.33$)  | 2.3% ($\pm 14.0\%$) |
>   | Random  | 0.0104 ($\pm 0.0023$) |  55.9\% ($\pm 9.7\%$)  |          -           |          -          |
>   | G2MILP  | 0.0073 ($\pm 0.0014$) | 69.1$\%$ ($\pm 5.9\%$) | 548.70 ($\pm 44.68$) | 19.3% ($\pm 6.6\%$) |
>
> - **Table 8 and Figure 2 show the same results.** We are sorry for the typo. The "Bowly" in the left figure in Figure 2 should be "Random". As we state in Section 4.2 III., on MIK,  instances generated by Bowly introduce numerical issues so that Ecole and SCIP fail to read them. We will correct the typo in a revised version.
>
> **Weakness 3.**
>
> > Code is not provided by authors.
>
> We will release our code once the paper is accepted to be published. We expect more studies on the topic of MILP instance generation, and our released code will be an important resource.

---

> > ### Comment · Reviewer_Vxmz · 2023-08-13
> > **Thank you. Increase score to Accept**
> >
> > I thank the authors for conducting additional experiments  in a short duration of time.( Especiall the downstream task expt which shows the capability of the model).
> > I am satisfied with the response to the question.
> >
> > I am increasing my score to accept.
> >
> > I hope the authors will release their code as promised( if the paper is accepted).

---

> > > ### Author Response · Authors · 2023-08-14
> > > **Thank you for your kind support.**
> > >
> > > Dear Reviewer Vxmz,
> > >
> > > Thanks for your kind support and for helping us improve the paper! We appreciate your valuable suggestions. We will release our code once the paper is accepted.
> > >
> > > Best,
> > >
> > > Authors

---

### Official Review · Reviewer_GoAh · 2023-07-06

**Soundness:** 3 good
**Presentation:** 3 good
**Contribution:** 3 good
**Rating:** 6
**Confidence:** 4

**Summary:**

The paper generates MIP instances with the help of VAEs. Experiment results show that the generated samples remain the characters of the training dataset from several aspects (e.g. the graph statistical value, the solving time of the generated samples, and the number of branching nodes of the generated samples), The evaluation shows that the generation helps the diversity, and preserves the difficulty. The samples may further help the NNs in downstream learning for MIP training, the authors introduces a task that predicts the optimal objective value, models trained on G2MILP shows obvious improvement on the testing dataset.

**Strengths:**

The task is highly valuable.
The idea of generating instances via deep neural networks is novel and sound. The whole framework provides an approach that generates new instances without the need to start form scratch.
Experiment results prove the effectiveness of the proposed method across various datasets (MIS, MIK, set covering and a nurse scheduling problem from MIPLIB2017).

**Weaknesses:**

N/A

**Questions:**

In the experiment results, the ratio $\eta$ is controlled as $0.01, 0.05$ and $0.1$. At most only 10% of the nodes in the original V-C bipartite graph are changed.

1.	Why doesn’t the generator set  larger $\eta$ for a complete comparison? (As the other baselines may generate the instances from scratch.) From the table, the effectiveness of the method quickly drops as $\eta$ increases.


**Limitations:**

As far as I could understand, the generator could only output the instances with the same problem scale.

---

> ### Author Rebuttal · Authors · 2023-08-09
>
> We thank the reviewer for the insightful and valuable comments. We respond to your comments as follows and sincerely hope that our rebuttal could properly address your concerns. If so, we would deeply appreciate it if you could raise your score and your confidence. If not, please let us know your further concerns, and we will continue actively responding to your comments and improving our submission.
>
> **Questions**
>
> > In the experiment results, the ratio $\eta$ is controlled as 0.01,0.05 and 0.1. At most only 10% of the nodes in the original V-C bipartite graph are changed.
> >
> > 1. Why doesn’t the generator set larger $\eta$ for a complete comparison? (As the other baselines may generate the instances from scratch.) From the table, the effectiveness of the method quickly drops as $\eta$ increases.
>
> - Thanks for your suggestion! **We have conduct ablation studies on the effect of $\eta$ on MIK.** The results are in **Figure 4** in the newly submitted PDF. The experimental settings are the same with those in Table 2 and Figure 2. From the figure we have the following conclusions.
>   - Though empirically a smaller $\eta$ leads to a relatively better performance, G2MILP maintains a high similarity performance even when $\eta$ is large.
>   - The downstream task performance does not drops significantly. This makes sense because smaller $\eta$ leads to more similar instances, while larger $\eta$ leads to more diverse (but still similar) instances, both of which can bnefit downstream tasks.
>   - G2MILP always outperforms Random, which demonstrates that the learning paradigm helps maintain the performance.
>   - Bowly fails on this dataset because its generated instances lead to numerical issues and cannot be read by Gurobi or SCIP.
>
> - Though Bowly generates instance from scratch, our results show that instances generated by Bowly are specious and unlikely to be realistic. This is because previous methods, including Bowly, are designed for research instead of developing solvers in real-world applications. Therefore, we can hardly use those methods in real-world scenarios. G2MILP, to the best of our knowledge, is the first method that can learn to generate realistic instances.
> - **Because our aim is data augmentation, it is unnecessary to generate instances from scratch.** As an analogy, G2SAT generates SAT instances by splitting existing SAT graphs into templates and then learning to merge them to form new formulas. This technique then develops as a standard manner in SAT generation area. We believe modifying existing instances will become an important line in developing better MILP generators.
>
> **Limitations**
>
> > The generator could only output the instances with the same problem scale.
>
> - Good point! **It is possible that G2MILP outputs instance with slightly different scales.** For example, we can add a virtual constraint, mask it and generate a new one to obtain an instance with one more constraint. Similarly, we can delete two constraints and generate a new one to obtain an instance with one less constraint. However, this method might seem trivial and unnecessary, because current implementation of G2MILP has brought good performance, and this method can hardly bring additional improvements.
> - **Generating MILP instances with different scales but similar semantic information and similar difficulty is a hard task.** Even in the community of SAT generation it has not been well solved. We recognize it as a valuable research direction and plan to study it in our future work.

---

### Official Review · Reviewer_qq1T · 2023-07-07

**Soundness:** 3 good
**Presentation:** 4 excellent
**Contribution:** 4 excellent
**Rating:** 7
**Confidence:** 5

**Summary:**

The paper addresses the challenge of limited availability of real-world mixed-integer linear programs (MILPs) instances for machine learning techniques employed in combinatorial optimization problems. The scarcity of these instances often leads to sub-optimal decisions and biased solver assessments. Existing solutions either depend heavily on expert-designed formulations or fail to capture the intricate features of real-world instances.

The authors propose G2MILP, the first deep generative framework for creating MILP instances. G2MILP treats MILP instances as bipartite graphs and uses a masked variational autoencoder to iteratively corrupt and replace parts of the original graphs, generating new ones. This approach allows G2MILP to learn to generate novel and realistic MILP instances without the need for expert-designed formulations, while simultaneously preserving the structures and computational hardness of real-world datasets. The generated instances can help improve MILP solvers when data availability is limited.

The authors also present a set of benchmarks to evaluate the quality of the generated MILP instances. The experimental results show that G2MILP is capable of producing instances that closely mimic real-world datasets in terms of both structures and computational hardness.

**Strengths:**

1. This is the first deep generative model for MILP instance generation based on the well known variant-constraint bipartite representation.

2. My favorite part of this work is the masking-based pre-training scheme, making me recall the masked language modeling objective in large language modeling. My feeling is that this direction could probably lead to large optimization model pre-training in the L2O field and this could be a very good start.

3. The empirical performance is good.

**Weaknesses:**

1. I have the feeling that the problem similarity metric is too privileged for the proposed method, as it essentially modifies existing problem instances.

2. The effectiveness seems to drop quickly with higher ratio of variable/constraint alternations.

**Questions:**

N/A

---

> ### Author Rebuttal · Authors · 2023-08-09
>
> We thank the reviewer for the positive and valuable comments. We respond to your comments as follows and sincerely hope that our rebuttal could properly address your concerns. If so, we would deeply appreciate it if you could raise your score. If not, please let us know your further concerns, and we will continue actively responding to your comments and improving our work.
>
> **Weakness 1.**
>
> > I have the feeling that the problem similarity metric is too privileged for the proposed method, as it essentially modifies existing problem instances.
>
> - **The used similarity benchmark is based on many previous works.** The considered statistics and features are commonly used in SAT generation [1] and MILP studies [2]. Measuring similarity by computing the distributional divergences is a common practice in benchmarking graph generation models [3].
> - **Beyond these similarity metrics, we also access the downstream task performance.** The results demonstrate that instances generated by G2MILP are not only realistic, but can also benefit real-world applications.
> - **Measuring MILP instance quality is a hard task. Currently no strong enough metrics have been developed, and we may open up the direction.** We believe that with the development of MILP generation techniques, studies on benchmarking MILP instances will follow up, and exploring MILP space will become possible. This is in line with the development law of generative model research.
> - **Modifying existing problem instances is not a trivial task, and G2MILP works well.** See the comparison with the Random baseline. Random also modifies existing instances, but generated instances are specious and unlikely to be realistic.
>
> - **Because our aim is data augmentation, it is unnecessary to generate instances from scratch.** As an analogy, G2SAT generates SAT instances by splitting existing SAT graphs into templates and then learning to merge them to form new formulas. This technique then develops as a standard manner in SAT generation area. We believe modifying existing instances will become an important line in developing better MILP generators.
>
> [1] Jiaxuan You, Haoze Wu, Clark Barrett, Raghuram Ramanujan, and Jure Leskovec. G2sat: learning to generate sat formulas. Advances in neural information processing systems, 32, 2019.
>
> [2] Frank Hutter, Lin Xu, Holger H Hoos, and Kevin Leyton-Brown. Algorithm runtime prediction: Methods & evaluation. Artificial Intelligence, 206:79–111, 2014.
>
> [3] Nathan Brown, Marco Fiscato, Marwin HS Segler, and Alain C Vaucher. Guacamol: benchmarking models for de novo molecular design. Journal of chemical information and modeling, 59(3):1096–1108, 2019.
>
> **Weakness 2.**
>
> > The effectiveness seems to drop quickly with higher ratio of variable/constraint alternations.
>
> **We have conduct ablation studies on the effect of the masking ratio $\eta$ on MIK.** The results are in **Figure 4** in the newly submitted PDF. The experimental settings are the same with those in Table 2 and Figure 2. From the figure we have the following conclusions.
>
> - Though empirically a smaller $\eta$ leads to a relatively better performance, G2MILP maintains a high similarity performance even when $\eta$ is large.
> - The downstream task performance does not drops significantly. This makes sense because smaller $\eta$ leads to more similar instances, while larger $\eta$ leads to more diverse (but still similar) instances, both of which can benefit downstream tasks.
> - G2MILP always outperforms Random, which demonstrates that the learning paradigm helps maintain the performance.
> - Bowly fails on this dataset because its generated instances lead to numerical issues and cannot be read by Gurobi or SCIP.
>
> Moreover, in real applications, it is reasonable and flexible to adjust the hyperparameter $\eta$ to achieve good performances in different scenarios.

---

### Official Review · Reviewer_sAxU · 2023-07-15

**Soundness:** 2 fair
**Presentation:** 2 fair
**Contribution:** 2 fair
**Rating:** 5
**Confidence:** 3

**Summary:**

This paper is focused on using machine learning techniques to address combinatorial optimization problems, particularly mixed-integer linear programs (MILPs). Particularly, this paper proposed G2MILP, which is a generative model for generating realistic MILP instances. Without prior expert knowledge, G2MILP can generate expressive instances preserving computational hardness of real-world datasets. To achieve this, G2MILP adopts a masked VAE approach to iteratively mask original graphs to generate new ones. Extensive evaluations are made to demonstrate the effectiveness of the proposed approach.  This paper shows how to adopt the well-established concepts (e.g., masked VAE) to the new data type (i.e., MILP instances).

**Strengths:**

The main strength of this paper is its originality in my eyes. It introduces a generative framework for generating realistic MILP instances. Though different generative models have been established for various types of data, little attention is paid for using generative models to generate MILP instances for improving the solvers.  To better develop training-based MILP solver, adequate training data is needed, and the proposed model can help.

**Weaknesses:**

1.

The contribution of the paper is unclear.  Though the paper introduces a new research direction of generating MILP instances, its technical contributions are unclear.  Representing MILP using bipartite graph and capturing features using GNN have been proposed by Gasse et al. The major technical contribution of this work is using masked VAE for expressive and realistic MILP instance generation, which I think is heuristic and lacks novelty.

2.

The methodology is not easy to follow. Adopting the concept of masked VAE to generate expressive MILP instances seems a cute idea. But the masking process is confusing to me.

2.1. How do you generate the masked graph G_tilde? What is the distribution of p_tilde(G_tilde|G)? From lines 165-168, the masking is performed by randomly selecting a constraint vertex. Does this mean p_tilde(G_tilde|G) is just a uniform distribution or something like that?

2.2. What’s the intuition of making the masked graph as a conditional variable for the decoder?  In masked VAE, the masked images are the input of the encoder. More explanations about the difference between the original masked VAE and the proposed pipeline for MILP instance generation are expected.

3.

The evaluations are insufficient to fully validate the proposed framework.

3.1. There is a lack of ablation study to show that the masking scheme helps improve complex graph generation as claimed in lines 58 – 60. Though there are evaluations considering different masking ratios (0.01, 0.05, 0.1), they are insufficient to help understand the proposed masking scheme. Does the order of masking constraint vertices matter in the generation process?

3.2. What’s the reconstruction for each graph component (i.e., bias, degree, etc)? How the graph features affect the new instance generation (e.g., the density of the graph)? I guess the new instance generation will be harder if the original graph is dense and complex?


3.3. The downstream task evaluation is important to demonstrate the benefits of the proposed method. However, this evaluation is inadequate. What is the mask ratio for the downstream task evaluation (figure 2)? I guess the performance will decrease if the mask ratio is high? Can you report the std for performance improvement given different sets of 20 generated instances? What about the performance on larger datasets? To which extend (e.g., the scarcity of the original dataset), the generated instances help improve the performance?

4.

The significance of the proposed method is unclear. Since the proposed method only mask constraint vertices, I think it is more like a graph generation pipeline. I am not fully convinced that we need to make this many efforts to generate new graphs. Furthermore, the benefits of new instances are not well demonstrated (as I question above).

5.

Presentations can be improved. The position of Table 2 needs to be re-arranged. Notation wise, I think it is a bit confusing to use the same G for both the input and the output of the model. Typo: there are two ‘and’s in line 67.


**Questions:**

I would like to adjust my rating if the authors can help clarify my confusions listed in the weakness section. Besides, I have other questions listed below:

1. Generating instances is not a new topic and G2SAT has explored this. Why it is non-trivial to adapt G2SAT to MILP instance generation? How does the consideration of the high-precision numerical prediction affect the design of the generative framework?

2. The generation of a new MILP instance is an iterative process. What is the computation cost for generating one MILP instance?



**Limitations:**

The limitations are discussed in the paper.

---

> ### Author Rebuttal · Authors · 2023-08-10
>
> We thank the reviewer for the insightful and constructive comments. We respond to your comments as follows and sincerely hope that our rebuttal could properly address your concerns. If so, we would deeply appreciate it if you could raise your score. If not, please let us know your further concerns, and we will continue actively responding to your comments and improving our submission.
>
> **Weakness 1.**
>
> Due to space limitation, we state the contribution of our work in respect of both task setting and technique in **Global Response 1**.
>
> **Weakness 2.**
>
> > 2.1. generate the masked graph
>
> - Yes,  $\tilde{p}(\tilde{G}|G)$ is like a uniform distribution. Specifically, we generate $\tilde{G}$ by sampling a constraint vertex $\tilde{v}$ from all constraint vertices $\mathcal{V}$ following a uniform distribution over these vertices, i.e., $\tilde{v}\sim U(\mathcal{V})$. Then we mask $\tilde{v}$ to generate $\tilde{G}$.
>
> > 2.2. difference with MAE.
>
> - We assume that you refer to masked auto-encoder (MAE) in computer vision. If not, please kindly provide the literature citation. Due to space limitation, the differences are in **Global Response 1**.
>
> **Weakness 3.**
>
> > 3.1. Masking scheme.
>
> - We conduct ablation studies on more different masking ratios, from 0.01 to 1.0, which may help understand the masking scheme. The results are in **Figure 4** in the newly submitted PDF.
> - We test different orders of masking constraint vertices, including uniformly sampling and sampling according to the vertex indices. Results are in **Table 11**. We find that **uniformly sampling achieves the best performance**. Sampling according to indices leads to a performance decrease, maybe because constraints that are indexed closely are relevant and lead to error accumulation. We think these results are interesting, and will study it in the future work.
>
> > 3.2. Generation
>
> - We are not sure whether we understand the question correctly. The generation process is determined by (i) the masked graph and (ii) the latent variables. During training, the latent variables encode the information of original graphs (graph features), so the decoder learns to reconstruct the original graph from the masked graph according to the latent variables. During inference, the model is decoder-only and the latent variables are randomly sampled. Different latent variables lead to different generated samples, so the decoder can randomly generate new diverse instances.
> - Dense and complex original graphs may lead to harder generation because graph representation learning is more challenging.
>
> > 3.3. Downstream task evaluation.
>
> - For the results in Figure 2, we use a mixture of $10$ instances from $\eta=0.01$ and $10$ instances from $\eta=0.05$ together (see Line 572 in Appendix B.4), because we find the mixture brings a slight performance improvement.
> - We conduct ablation studies on masking ratio. Results are in **Figure 4(b)**. According to the results, though empirically a small masking ratio brings better performance, the performance does not decrease quickly with an increasing masking ratio.
> - We report the standard error for performance improvement in **Table 12**.
> - We conduct experiments on different sizes of the original datasets, as well as the ratio of generated instances to original ones, on MIS. The results are in **Table 13**. The results show that G2MILP can bring performance improvements on varying sizes of datasets.
>
> **Weakness 4.**
>
> - In real applications, low data availability is often a key bottleneck for solver development. Generating realistic instances is orthogonal to previous methods that directly focus on solver development, and can be applied in many different real-world scenarios. It does not take many efforts but can bring additional performance improvements.
> - G2SAT has attracted much attention since its publication, which implies that the graph generation pipeline for instance generation task is a proper topic for research community.
> - To further demonstrate the benefits of generated instances, we conduct an additional downstream task of a predict-and-search framework. Results are in **Table 14** and details are in **Global Response 3**.
>
> **Weakness 5.**
>
> - Thanks for your suggestions! We will re-arrange Table 2 and fix typos in a revised version. Notation wise, we will use $\hat{G}$ to denote the outputs for clarity.
>
> **Question 1.**
>
> - **G2SAT cannot determine numerical values of the coefficients in MILPs.** Specifically, SAT instances can be modeled as homogeneous bipartite graph (i.e., graphs without edge weights or node features), and G2SAT splits the graphs into fragments and then merges the fragments together to form new ones. This method can generate new topological structures of bipartite graphs, but when merging the fragments together, it is nontrivial how to determine the edge weights, i.e., the coefficient values.
> - **Even without numerical coefficients, G2SAT can hardly preserve reality of instances.** To see this, we conduct **an additional experiment** that transfers G2SAT to a special MILP dataset, MIS, in which all coefficients are 1.0 and thus the instances can be modeled as homogeneous bipartite graphs. We apply G2SAT to learn to generate new graphs and convert them to MILPs. Results are in **Table 9** in the newly submitted PDF. The results show that G2MILP significantly outperforms G2SAT on the special cases.
>
> **Question 2.**
>
> - We generate $1000$ instances using G2MILP with $100$ iterations for each instances. **The average time costs for generating one instance for MIS, SetCover, and MIK are 0.45s, 0.97s and 0.85s respectively.** The time cost is linearly related to the number of iterations. Though being an iterative process, G2MILP is fast because each iteration is very fast. As a comparison, our experiments on MIS (mentioned in response to **Question 1**) show that G2SAT takes about 47s to generate one instance.

---

> > ### Comment · Reviewer_sAxU · 2023-08-14
> > **Thank you. Increasing my score.**
> >
> > I appreciate the authors' time and efforts in carefully responding to each of my questions. The significance of the proposed method is clarified. Lots of additional evaluations are conducted and they are helpful in further demonstrating the effectiveness of the proposed approach. Overall, I like the originality of this work as I stated in the Strengths section.  Since my concerns and questions are all well addressed, I would love to increase my score.

---

> > > ### Author Response · Authors · 2023-08-14
> > > **Thank you for your kind support.**
> > >
> > > Dear Reviewer sAxU,
> > >
> > > Thanks for your kind support and for helping us improve the paper! We appreciate your valuable suggestions.
> > >
> > > Best,
> > >
> > > Authors

---

### Official Review · Reviewer_K4a3 · 2023-07-17

**Soundness:** 3 good
**Presentation:** 2 fair
**Contribution:** 2 fair
**Rating:** 5
**Confidence:** 3

**Summary:**

The paper proposed a deep generative framework G2MILP based on a masked variational autoencoder (VAE) for mixed-integer linear programs (MILP) instances. The proposed framework G2MILP corrupts and replaces constraints to generate new MILP instances while keeping the variables unchanged. Experiments show that G2MILP can produce instances that closely resemble real-world datasets in terms of both structures and computational hardness. Meanwhile, the generated instances from G2MILP can be used to augment the small training dataset to improve the prediction performance.

**Strengths:**

1. The paper is well motivated by the limited data issue of MILP in the real world and the shortage of a flexible generative framework that can generate high-quality MILP instances.
2. The proposed decoder consisting of four predictive modules seems quite tailor-designed for the problem.

**Weaknesses:**

1. The motivation of this work is similar to those Graph generation works and G2SAT, though targeting a different specific application domain. The main technique of masked VAE has already been well studied in the literature. So, this work is not that novel and interesting.
2. The proposed G2MILP can only modify the constraints of MILP beyond the training samples, which limits the contribution of this work. Although the authors claimed that their framework can be extended to change the variables of MILP, it is expected such flexibility can be achieved in this work.
3. Eq. (9) and Eq. (11) are inconsistent with Fig. 1. The logits/weights predictor shown in Fig. 1 is conditioned on $d_\tilde{v}$ but Eq. (9) and (11) are not. Moreover, from my understanding, the logits/weights prediction of $\delta_{\tilde{v}, u}$/$e_{u, \tilde{v}}$ should also be conditioned on $h_\tilde{v}$ and $z_{\tilde{v}}$.
4. The writing needs to be further improved. See the questions below.

**Questions:**

1. How does G2MILP deal with different n and m?  Different instances would have different n and m.
2. For the proposed normalization, would it be problematic when there is outlier in the dataset?
3. In Section 4.1, about structural distributional similarity, statistics that remain constant are excluded. Does it mean the similarity is only calculated in terms of the constraints?
4. For the Bowly baseline, can it change variables? If can, the similarity comparison may be unfair since the Bowly baseline introduces more diversity to the generated instances.
5. Good generation should introduce novelty compared to the training set, implying that the generated samples should differ from the training samples. So, it is tricky to use the structural distribution similarity to denote a good generation. I expect a discussion on this. t-SNE visualization for the baselines should be included.

**Limitations:**

The authors have discussed the limitations of their work, which however I think should be a part of this work (point 2 in the weakness).

---

> ### Author Rebuttal · Authors · 2023-08-09
>
> We thank the reviewer for the insightful and valuable comments. We respond to your comments as follows and sincerely hope that our rebuttal could properly address your concerns. If so, we would deeply appreciate it if you could raise your score. If not, please let us know your further concerns, and we will continue actively responding to your comments and improving the work.
>
> **Weakness 1.**
>
> - We state the novelty of our work in respect of both task setting and technique in **Global Response 1**. Due to space limitation, we summarize the statement here.
>
>   - Generating MILPs is much harder than generating SATs because it involves precise numerical prediction. G2SAT can hardly transfer to MILP generation.
>
>   - We assume that you refer to masked auto-encoder (MAE) developed in computer vision. If not, please kindly provide the literature citation. However, they are different because MAE is an auto-regressive method which aims to reconstruct given samples, while our masked VAE framework aims to generate new instances from masked ones.
>
> **Weakness 2.**
>
> - Thanks for your suggestion! **We have implemented a new version of G2MILP that supports modifying variables.** Results are in **Table 10** in the newly submitted PDF. Due to space limitation, you can refer to **Global Response 4** for more details.
>
> **Weakness 3.**
>
> > Eq. (9) and Eq. (11) are inconsistent with Fig. 1.
>
> - Thanks for your comment! **Fig 1 shows the inference process while Eq. (9) and Eq. (11) details the network implementation.** As shown in Fig. 1, the logits and weights prediction are conditioned on the degree. As for the detailed implementation, we use a NN to predict $\hat{d}_{\tilde{v}}$.
> We use another NN to predict the ${\hat{\delta}} _{\tilde{v},u}$ for each variable vertex $u$, and this NN is not conditioned on  $\hat{d} _{\tilde{v}}$ . Then the connected vertices are the $\hat{d} _{\tilde{v}}$ variable vertices with top $\hat{\delta} _{\tilde{v},u}$, and so the connected vertices are conditioned on both  $\hat{d} _{\tilde{v}}$  and  $\hat{\delta} _{\tilde{v},u}$ . We will improve the presentation in a revised version to avoid inconsistency.
>
> > The logits/weights prediction should be conditioned on $ℎ_{\tilde{v}}$ and $z_{\tilde{v}} $.
>
> - Thanks for the insightful question! As we add virtual edges between $\tilde{v}$ and all variable nodes (Line 167), there should be message passing between $\tilde{v}$ and the variable nodes, and thus the information of $\tilde{v}$ has been encoded into other node representations. Therefore, we do not explicitly use $h_{\tilde{v}}$ and $z_{\tilde{v}}$ as inputs for logits/weights prediction. Actually we have tried such implementation before, but found that it introduced additional parameters while the impact on performance was limited.
>
> **Weakness 4.**
>
> - We respond to your questions as below, and we will polish our writing in a revised version to make the questions clear in our paper.
>
> **Questions 1.**
>
> - For the **inputs**, G2MILP allows any $n$ and $m$ because the model does not depends on instance size. For the **outputs**, current implementation of G2MILP does not change $n$ and $m$ of input instances.
>
> - It is possible that G2MILP outputs instances with slightly different $n$ and $m$. For example, we can add a virtual constraint, mask it and generate a new one to obtain an instance with one more constraint. Generating MILP instances with different scales but similar semantic information and similar difficulty is an interesting topic, and we plan to study it in our future work.
>
> **Question 2.**
>
> > For the proposed normalization, would it be problematic when there is outlier in the dataset?
>
> - In real applications, instances in the training set are from real-world scenarios. Thus covering all such cases is meaningful.
> - Every model might be impacted by outlier data. However, the normalization makes the model more robust to outlier cases, because it maps all values to $[0,1]$, which makes it easier for the model to learn.
> - Empirically, we do not find obvious outliers in several datasets.
>
> **Question 3.**
>
> - **We consider comprehensive metrics including coefficient density, constraint and variable degrees, coefficient means, et al.** We excluded statistics such as problem sizes, numbers of  integer variables, et al. For descriptions of the considered statistics, readers can refer to Table 7 in Appendix B.3.
>
> **Question 4.**
>
> - **Bowly  generates instances from scratch by randomly sampling the coefficients.** It was not designed for learning datasets to generate realistic instances, and we can only contral simple statistics like problem sizes and coefficient means. For a fair comparison, we set all controllable parameters  the same as the training sets.
> - **We do not aim to show that G2MILP is better than Bowly in every metrics, because the two methods are for different tasks.** Our results show that instances generated by Bowly are unlikely to be realistic. In other word, it cannot learn to generate data that is iid. with existing data. Therefore, we can hardly use it in low-resource MILP solver development, which is also verified by downstream task experiments.
>
> **Question 5.**
>
> > Good generation should introduce novelty compared to the training set.
> >
> > It is tricky to use the structural distribution similarity to denote a good generation.
>
> - We state the rationality of our evaluation metrics in **Global Response 2**. Due to space limitation, we summarize the response as follows.
>   - The used similarity benchmark is based on many previous works.
>   - Beyond these similarity metrics, we also access the downstream task performance.
>   - Currently no strong enough metrics have been developed.
>
> > t-SNE visualization for the baselines should be included.
>
> - The t-SNE visualization for baselines are in **Figure 5** in the newly submitted PDF. G2MILP generates diverse instances around the training set, while instances generated by Random are more biased from the realistic ones.

---

> > ### Comment · Reviewer_K4a3 · 2023-08-17
> > **Concerns have not been fully resolved.**
> >
> > I appreciate authors’ efforts in addressing my concerns. However, they have not been fully resolved:
> >
> > 1. Beyond MAE developed in computer vision, I would more like to emphasize that masked auto-encoder has already explored widely in graphs, e.g., [1, 2, 3]. What I want to stress is that the mask idea for reconstruction/generation has been well explored in the community. In addition, there is also masked VAE in graphs [4]. While I acknowledge that there is some novelty in this work, a Taylor-designed masked VAE for a new specific application, I don't think the idea of the paper is that interesting or that new given the literature.
> >
> > 2. For weakness 3, I am confused what you explained. $p_\theta(\delta_{\hat{v}, u}|G, Z, d_\hat{v})$ in Fig. 1 describes the condition on $d_\hat{v}$, but Eq. (9) is not.
> >
> > 3. For question 1, how does G2MILP deal with different $n$ and $m$ of the inputs? For the output, from my understanding, G2MILP would need to deal with different $n$ and $m$ for different instances. If I am wrong, please correct me.
> >
> > 4. My question 3 indeed challenges the rationality of current structural similarity. Excluding the statistics of variables is unreasonable, as these statistics form a crucial part of MILP instances. This exclusion also leads to the inability to evaluate “v” models mentioned in Table 10. I think it is more reasonable to consider statistics of variables when calculating the structural similarity.
> >
> > References:
> >
> > [1] Hou, Z., Liu, X., Cen, Y., Dong, Y., Yang, H., Wang, C., & Tang, J. (2022, August). Graphmae: Self-supervised masked graph autoencoders. In Proceedings of the 28th ACM SIGKDD Conference on Knowledge Discovery and Data Mining (pp. 594-604).
> >
> > [2] Tan, Q., Liu, N., Huang, X., Choi, S. H., Li, L., Chen, R., & Hu, X. (2023, February). S2GAE: Self-Supervised Graph Autoencoders are Generalizable Learners with Graph Masking. In Proceedings of the Sixteenth ACM International Conference on Web Search and Data Mining (pp. 787-795).
> >
> > [3] Tan, Q., Liu, N., Huang, X., Chen, R., Choi, S. H., & Hu, X. (2022). Mgae: Masked autoencoders for self-supervised learning on graphs. arXiv preprint arXiv:2201.02534.
> >
> > [4] Li, X., Ye, T., Shan, C., Li, D., & Gao, M. (2023, April). SeeGera: Self-supervised Semi-implicit Graph Variational Auto-encoders with Masking. In Proceedings of the ACM Web Conference 2023 (pp. 143-153).

---

> > > ### Author Response · Authors · 2023-08-18
> > > **Response to the concerns (1/2)**
> > >
> > > Dear Reviewer K4a3,
> > >
> > > Thanks for your reply and for your valuable comments! We are glad to respond to your concerns as follows.
> > >
> > > > 1. Beyond MAE developed in computer vision, I would more like to emphasize that masked auto-encoder has already explored widely in graphs, e.g., [1, 2, 3]. What I want to stress is that the mask idea for reconstruction/generation has been well explored in the community. In addition, there is also masked VAE in graphs [4]. While I acknowledge that there is some novelty in this work, a Taylor-designed masked VAE for a new specific application, I don't think the idea of the paper is that interesting or that new given the literature.
> > >
> > > Thanks for kindly providing the valuable references. We appreciate your feedback, and we would like to further emphasize our technical contribution in light of the existing literature. We will incorporate the citations of the relevant papers and discuss them in the revised version.
> > >
> > > - **Our main contribution lies in identifying MILP generation as a valuable task and proposing the first framework as a feasible solution.** MILP plays a crucial role in combinatorial optimization research and finds wide application in various industrial optimization scenarios, while the low data availability is a bottleneck challenge in developing powerful MILP solvers. **Therefore, we believe that our work is aligned with current trends and will be significant in promoting the advancement of the MILP solver area.** While our work builds upon existing techniques, we have carefully designed our method specifically for this new task, demonstrating its effectiveness.
> > >
> > > - **Though the ideas of MAE and VAE have been extensively explored in the literature, developing new variants for novel applications remains valuable.** MAE and VAE serve as foundational models that have inspired numerous studies across diverse domains. Designing tailored model structures to suit specific scenarios is challenging and has motivated significant research efforts. In the context of MILP generation, we carefully design the mask-and-generate mechanism to preserve realism, and we design the decoder consisting of four predictive modules specifically for the bipartite graph representation of MILPs.
> > >
> > > - **Differences from [1, 2, 3].** These three papers explore the use of MAE for self-supervised learning on graphs and are relevant to our work. Our method is significantly different from them, and the reasons are similar to the differences we have highlighted regarding MAE in **Global Response 1**.
> > >
> > > - **Differences from [4].** The paper [4] presents a graph VAE with masking for graph self-supervised learning. Our method is different from [4] in several aspects.
> > >
> > >   - **Motivation:**  While [4] focuses on generative graph self-supervised learning with VAE, our focus is on generating new graphs.
> > >   - **Use of masking:** [4] employs masking in a VAE model only for data augmentation. In contrast, our aim is to generate new graphs from masked input graphs.
> > >   - **Model Structure:** In [4], the masked graphs are inputs to the encoder for data augmentation. In contrast, in our work, the masked graphs are inputs to the decoder because we aim to generate new graphs from masked ones.
> > >   - **Theory:** The aforementioned differences result in distinct derivations of the evidence lower bound (ELBO) in our work compared to [4].
> > >
> > > > 2. For weakness 3, I am confused what you explained. $p_\theta(\delta_{\tilde{v},u}|\tilde{G},Z,d_{\tilde{v}})$ in Fig. 1 describes the condition on $d_{\hat{v}}$, but Eq. (9) is not.
> > >
> > > - In Fig.1, the probabilities $p_\theta(\delta_{\tilde{v},u}|\tilde{G},Z,d_{\tilde{v}})$ correspond to the **true logits** $\delta_{\tilde{v},u}$ (taking values in $\{0,1\}$), which indicate whether two vertices $\hat{v}$ and $u$ are connected. These probabilities are conditioned on the degrees $d_{\tilde{v}}$ in our probability modeling.
> > > - On the other hand, in Eq. (9), we describe the output of the neural network MLP$_\theta^{logits}$, which provides the **predicted logits values** $\hat{\delta} _{\tilde{v},u}$ (taking values in the range $(0,1)$) that indicate the likelihood of the connection between the two vertices. The network does not take the predicted degrees as inputs.
> > > - As we state in Line 202, we connect $\hat{d} _{\tilde{v}}$ variable vertices with top logits. To be precise, the selected vertices to connect are those $u\in argTopK(\{\hat{\delta} _{\tilde{v},u}|u\}, \hat{d} _{\tilde{v}})$ (also see Algorithm 2 in Appendix A.3.3). Therefore, even though $\hat{d} _{\tilde{v}}$ is not an input to the neural network in Eq. (9), the decision of whether a vertex $u$ is linked to $\tilde{v}$ is conditioned on both $\hat{\delta} _{\tilde{v},u}$ and $\hat{d} _{\tilde{v}}$.
> > > - If the reviewer still finds the notation confusing, we will refine the notation to make it more clear and unambiguous in the revised version. We appreciate the reviewer's input and will address this concern accordingly.

---

> > > > ### Author Response · Authors · 2023-08-18
> > > > **Response to the concerns (2/2)**
> > > >
> > > >
> > > >
> > > > > 3. For question 1, how does G2MILP deal with different $n$ and $m$ of the inputs? For the output, from my understanding, G2MILP would need to deal with different $n$ and $m$ for different instances. If I am wrong, please correct me.
> > > >
> > > > - **For inputs.** We represent each MILP instance as a bipartite graph, where $n$ corresponds to the number of variable vertices, and $m$ corresponds to the number of constraint vertices. Therefore, different values of $n$ and $m$ indicate varying sizes of bipartite graphs. Note that the message passing-based GNNs can handle graphs with different sizes, because they use local information propagation and aggregation mechanisms to capture node features, which are independent of the graph sizes. Readers can refer to Appendix A.3.1 for the GNN architecture. Because G2MILP leverages GNNs to deal with input graphs, it can naturally deal with different $n$ and $m$.
> > > >
> > > > - **For outputs.** G2MILP generates a new instance $G'$ by modifying vertices of an existing graph $G$. Since the output $G'$ is derived from the input $G$, it will have the same $n$ and $m$ as the input. As different input instances $G$ have different $n$ and $m$, the corresponding outputs $G'$ will naturally have different $n$ and $m$.
> > > >
> > > >
> > > >
> > > > > 4. My question 3 indeed challenges the rationality of current structural similarity. Excluding the statistics of variables is unreasonable, as these statistics form a crucial part of MILP instances. This exclusion also leads to the inability to evaluate “v” models mentioned in Table 10. I think it is more reasonable to consider statistics of variables when calculating the structural similarity.
> > > >
> > > > - Thanks for your suggestion. First we want to point out that our considered metrics have included statistics that are relevant to variables, e.g., variable degrees and coefficient density.
> > > > - **We value your suggestion and incorporate three additional statistics in the computing of similarity scores: (1) mean of objective coefficients $\mathbf{c}$, (2) std of objective coefficients $\mathbf{c}$, and (3) the ratio of continuous variables.** With these additional metrics, we have recomputed the  structural similarity scores and updated the results in both **Table 2** and **Table 10**. The new results are as follows. From the results, we can still conclude that G2MILP outperforms all baselines, further supporting the effectiveness of our proposed method.
> > > >
> > > > **Table 2 recomputed.** Structural distributional similarity scores between the generated instances with the training datasets. Higher is better. η is the masking ratio. We do not report the results of Bowly on MIK because Ecole and SCIP fail to read the generated instances due to large numerical values.
> > > >
> > > > |                      | MIS   | SetCover | MIK   |
> > > > | -------------------- | ----- | -------- | ----- |
> > > > | Bowly                | 0.144 | 0.150    | -     |
> > > > | Random ($\eta=0.01$) | 0.722 | 0.791    | 0.971 |
> > > > | G2MILP ($\eta=0.01$) | 0.997 | 0.874    | 0.994 |
> > > > | Random ($\eta=0.05$) | 0.670 | 0.704    | 0.878 |
> > > > | G2MILP ($\eta=0.05$) | 0.951 | 0.833    | 0.969 |
> > > > | Random ($\eta=0.1$)  | 0.618 | 0.648    | 0.768 |
> > > > | G2MILP ($\eta=0.1$)  | 0.921 | 0.834    | 0.930 |
> > > >
> > > > **Table 10 recomputed.**  Results of different implementations of G2MILP on MIK. In the table, $\eta$ denotes mask ratio, “v” denotes only modifying variables (objective coefficients and variable types), “c” denotes only modifying constraints, and “vc” denotes first modifying variables and then modifying constraints.
> > > >
> > > > |                      | v     | c     | vc    |
> > > > | -------------------- | ----- | ----- | ----- |
> > > > | G2MILP ($\eta=0.01$) | 0.998 | 0.988 | 0.985 |
> > > > | G2MILP ($\eta=0.05$) | 0.996 | 0.968 | 0.967 |
> > > > | G2MILP ($\eta=0.1$)  | 0.996 | 0.928 | 0.912 |

---

> > > > > ### Comment · Reviewer_K4a3 · 2023-08-21
> > > > > **Thanks for the responses.**
> > > > >
> > > > > I appreciate authors' efforts in addressing my concerns. Most of them have been carefully addressed.
> > > > >
> > > > > 1. I can understand the differences and the contribution to the specific domain that authors pointed out. But I still maintain my opinion regarding the novelty of this paper. The utilization of a generative framework has been a well-established concept in the literature. The main technique of masked (V)AE already been well studied in the literature. I am not saying the paper is not novel at all, but not that novel.
> > > > >
> > > > > 2. I do not agree the explanation for weakness 3. $p_\theta(\delta_{\tilde{v}, u} | \tilde{G}, Z, d_{\tilde{v}})$ describes the probability form of the logit predictor MLP_{\theta}. Therefore, the implementation of the neural network should be consistent with it. The described connection pertains to how the predicted probability is utilized.
> > > > >
> > > > > Given these, I increase my score to borderline accept.

---

> > > > > > ### Author Response · Authors · 2023-08-21
> > > > > > **Thank you for your kind support.**
> > > > > >
> > > > > > Dear Reviewer K4a3,
> > > > > >
> > > > > > Thanks for your kind support and for helping us improve the paper! We appreciate your valuable suggestions.
> > > > > >
> > > > > > 1. We sincerely appreciate your consideration of the technical differences and the contribution we make to the domain. It is encouraging to know that our efforts in addressing your concerns have been well-received.
> > > > > >
> > > > > > 2. Thanks for your suggestion. In the final version, we will improve the notation for better consistency throughout the paper. Specifically, we will revise the notations as:
> > > > > >
> > > > > >    $$\hat{\delta}' _{\tilde{v},u}=\sigma(\text{MLP} _{\theta}^{logits}([\mathbf{h} _u,\mathbf{z} _u])),$$
> > > > > >
> > > > > >    and
> > > > > >
> > > > > >     $$ \hat{\delta} _{\tilde{v},u} =   \begin{cases} 1,  u\in argTopK(\\{ \hat{\delta}' _{\tilde{v},u}|u\in\mathcal{W} \\},\hat{d} _{\tilde{v}} ), \\\\ 0, \text{otherwise,} \end{cases}$$
> > > > > >
> > > > > >
> > > > > >    so that $\hat{\delta} _{\tilde{v},u}$ is conditioned on $\hat{d} _{\tilde{v}}$, aligning Eq (9) with Fig. 1.
> > > > > >
> > > > > > Best,
> > > > > >
> > > > > > Authors

---

> > > ### Author Response · Authors · 2023-08-21
> > > **It is our pleasure to discuss with you.**
> > >
> > > Dear Reviewer K4a3,
> > >
> > > We deeply appreciate your time and insightful comments. We sincerely hope that our response could properly address your concerns. If you have any further concerns, we are more than happy to discuss with you and keep improving this work.
> > >
> > > Best,
> > >
> > > Authors

---

### Author Rebuttal · Authors · 2023-08-10

Dear reviewers,

We sincerely thank all reviewers' insightful and constructive comments, which helped to significantly improve our work. We respond to the comments given by each reviewer in detail, and in this global response,  we answer several key issues raised by multiple reviewers.

**1. Contribution & Novelty**

- **Task setting.**

  **MILP generation is much harder than SAT generation because it involves not only topological structure prediction, but also precise numerical prediction.** Specifically, SAT instances can be modeled as homogeneous bipartite graph (i.e., graphs without edge or node features), so current SAT generation techniques merge graph templates together to form new ones. These methods can hardly adapt to MILP generation tasks because they cannot determine the coefficients.

  Nonetheless, we conduct **an additional experiment** that transfers G2SAT to a special MILP dataset, MIS, in which all coefficients are 1.0 and thus the instances can be modeled as homogeneous bipartite graphs. We apply G2SAT to learn to generate new graphs and convert them to MILPs. Results are in **Table 9** in the newly submitted PDF. The results show that G2MILP significantly outperforms G2SAT on the special cases.

- **Technique.**

  **Our proposed masked VAE is novel and different from previous masked auto-encoder (MAE) in computer vision.**

  - **Motivation:** MAE is an **auto-regression** method that learns to reconstruct the masked images for **representation learning**. Masked VAE, in contrast, is a **VAE** based **generative model** that aims to generate new diverse instances---instead of reconstructing the original instances---from a masked one.
  - **Model Structure:** Masked VAE has a resample layer (Eq. 5) as well as a prior loss (Eq. 4) when training, and works in a decoder-only manner when inference. For decoder-only generation, masked VAE takes masked graphs as inputs of the decoder instead of the encoder, which is also different from MAE.
  - **Theory:** Moreover, we theoretically derive the evidence lower bound (ELOB) to incorporate the mask process (Eq. 14), which is an extension of VAE theory.

**2. Evaluation Metrics.**

- **The used similarity benchmark is based on many previous works.** The considered statistics and features are commonly used in SAT generation [1] and MILP studies [2]. Measuring similarity by computing the distributional divergences is a common practice in benchmarking graph generation models [3].
- **Beyond these similarity metrics, we also access the downstream task performance.** The results demonstrate that instances generated by G2MILP are not only realistic, but can also benefit real-world applications. Novelty and diversity are also metrics that measure generative model performance. However, these two metrics are never easy to define for MILP instances. The downstream task performance improvements imply that instances generated by MILP have sufficient novelty and diversity for real applications.
- **Measuring MILP instance quality is a hard task. Currently no strong enough metrics have been developed, and we may open up the direction.** We believe that with the development of MILP generation techniques, studies on benchmarking MILP instances will follow up, and exploring MILP space will become possible. This is in line with the development law of generative model research.

**3. We conduct an additional downstream task for a neural solver, i.e., the predict-and-search framework proposed in [4].** Specifically, [4] proposes a framework that first predicts a solution and then uses solvers to search for the optimal solutions in a trust region. We consider using generated instances to enhance the predictive model. We first train the predictive model on 100 MIS instances, and then use the generative models to generate 100 new instances to augment the dataset. The results are in **Table 14**. Bowly generates low-quality data that disturbs the model training, so that there is no optimal solution in the trust region around the predicted solution. Though both Random and G2MILP can enhance the solving framework to reduce solving time, we can see G2MILP significantly outperforms Random.

**4.  We implement a new version of G2MILP that supports modifying variables.** Specifically, in each iteration, we randomly mask a variable and use new modules to generate the objective coefficient and the variable type. We fine-tune the previous model trained on MIK to obtain a new model with a variable generation module, and test its performance on MIK. We conduct experiments with different implementations, and the results are in **Table 10** in the newly submitted PDF. In future work, we plan to implement more kinds of modifying operators to generated more diverse MILP instances.

**5. We conduct extensive experimental analysis.**

- Different orders of masking constraints. Results are in **Table 11**.
- Performance improvements on different sizes of datasets and ratio of generated instances. Results are in **Table 13**.
- Impact of masking ratio $\eta$ on similarity scores and downstream task performance improvements. Results are in **Figure 4**.
- t-SNE visualization for baselines. See **Figure 5**.



[1] Jiaxuan You, Haoze Wu, Clark Barrett, Raghuram Ramanujan, and Jure Leskovec. G2sat: learning to generate sat formulas. Advances in neural information processing systems, 32, 2019.

[2] Frank Hutter, Lin Xu, Holger H Hoos, and Kevin Leyton-Brown. Algorithm runtime prediction: Methods & evaluation. Artificial Intelligence, 206:79–111, 2014.

[3] Nathan Brown, Marco Fiscato, Marwin HS Segler, and Alain C Vaucher. Guacamol: benchmarking models for de novo molecular design. Journal of chemical information and modeling, 59(3):1096–1108, 2019.

[4] Qingyu Han, Linxin Yang, Qian Chen, Xiang Zhou, Dong Zhang, Akang Wang, Ruoyu Sun, 410 and Xiaodong Luo. A gnn-guided predict-and-search framework for mixed-integer linear 411 programming. ICLR, 2023.

---

### Decision · Program_Chairs · 2023-09-21

**Decision:**

Accept (spotlight)

**Comment:**

The paper proposes a deep generative model for MILP instances. The reviews agree that this novel and exciting use of generative models. Some concern was raised about the technical novelty in the underlying technology; the use of Autoencoders on graphs. However, in this case, the application domain can be argued to provide sufficient novelty. After the rebuttal, where an additional baseline and task are added, and analysis conducted, the paper received unanimous accept recommendations across 6 reviewers. This is an exciting direction for machine learning, and I am happy to recommend acceptance.